

# Numerical modeling and sensitivity analysis of seawater intrusion in a dual-permeability coastal karst aquifer with

**conduit networks**

Zexuan Xu[1, *], Bill X. Hu[2] and Ming Ye[3]

[1]Climate and Ecosystem Sciences Division, Lawrence Berkeley National Laboratory,

Berkeley, California, 94720, USA

[2]Institute of Groundwater and Earth Sciences, Jinan University, Guangzhou, Guangdong,

China

[3]Department of Scientific Computing, Florida State University, Tallahassee, Florida,

32306, USA

*Corresponding author: email address: xuzexuan@gmail.com;

Submitted to Hydrology and Earth System Sciences



**Abstract**

In a coastal karst aquifer, seawater intrudes significantly landward through the
highly permeable subsurface conduit system, and contaminates the groundwater
resources in the porous medium. In this study, a two-dimensional coupled density-
dependent flow and transport SEAWAT model is developed to study seawater intrusion
in the dual-permeability karst aquifer. To provide guideline for modeling seawater
intrusion in such an aquifer, local and global sensitivity analysis are conducted to
evaluate the parameters of boundary conditions and hydrological characteristics,
including hydraulic conductivity, effective porosity, specific storage and dispersivity of
the conduit and the porous medium. In the local sensitivity analysis, simulations are more
sensitive to all parameters at the seawater and freshwater mixing zone than elsewhere.
The most important parameter for simulations in both domains is salinity at the
submarine spring, which is also the boundary condition of the conduit. The hydrological
characteristics of the conduit network are not only important to the simulations in the
conduit, but also significantly affect the simulations in the porous medium, due to the
interactions between the two systems. Therefore, salinity and head observations in the
conduits and karst features have more values for calibrating the models and
understanding seawater intrusion in a coastal karst aquifer. Compared to the local
sensitivity analysis, the global sensitivity results are different in several parameters
(hydraulic conductivity, porosity and the boundary conditions at the submarine spring),
mainly due to the non-linear relationship of the parameters with respect to the
simulations. The results of global sensitivity analysis also indicate that the Darcy's
equation does not accurately calculate the conduit flow rate with hydraulic conductivity





in the continuum SEAWAT model. Dispersivity is no longer an important parameter in

45     the advection-dominated transport aquifer system with conduit, compared to the

sensitivity results in a homogeneous porous medium. Based on the sensitivity analysis,

the extents of seawater intrusion are quantitatively evaluated with the identified important

parameters, including salinity at the submarine spring with rainfall recharge, sea level rise

and longer simulation time under an extended low rainfall period.

Key Words: Seawater intrusion; Coastal karst aquifer; Variable-density numerical model;

Dual-permeability karst system; Sensitivity analysis

## 1. Introduction

Many serious environmental issues have been caused by seawater intrusion in the

coastal regions, such as soil salinization, marine and estuarine ecological changes, and

groundwater contamination (Bear, 1999). Groundwater salinization is the primary

detrimental effect of seawater intrusion to groundwater resources, since mixing with less

than 1% of seawater (250 mg/L chloride) by volume makes freshwater non-potable

(WHO, 2011). Custodio (1987) and Shoemaker (2004) summarized the controlling

factors of seawater intrusion into a coastal aquifer, including the geologic and lithological

heterogeneity, localized surface recharge, paleo-hydrogeological conditions and

anthropogenic influences. Werner et al. (2013) concluded that climate variations,

groundwater pumping, and fluctuating sea levels are important to the mixing of seawater

and freshwater. On the other hand, sea level rise has been recognized as a serious

environmental threat in this century (Voss and Souza, 1987; Bear, 1999; IPCC, 2007).



Sea level fluctuation has significant and complex impacts on seawater intrusion in a coastal aquifer under different conditions (Werner et al., 2013). The Ghyben-Herzberg relationship indicates that rising sea level causes extended seawater intrusion in a coastal aquifer, and significantly moves the mixing interface position further landward (Werner and Simmons, 2009). Essink et al. (2010) pointed out that seawater intrusion is exacerbated under sea level rise conditions in a large time scale due to global climate change. Likewise, high tides associated with hurricanes or tropical storms have been found to temporarily affect the extent of seawater intrusion in a coastal aquifer (Moore and Wilson, 2005; Wilson et al., 2011). However, tide fluctuation is generally negligible to mixing interface movement over a regional and large time scale (Inouchi et al., 1990).

Modeling seawater intrusion in a coastal aquifer requires a coupled density-dependent flow and transport numerical model. In such model, the solution of seawater is based on the groundwater velocity field from flow modeling, and salinity in turn determines water density and affects the simulation of flow field. Several variable-density numerical models have been developed and widely used, including SUTRA (Voss and Provost, 1984) and FEFLOW (Diersch, 2002). SEAWAT is a widely used density-dependent model, which solves flow equations by finite difference method, and transport equations by three major classes of numerical techniques (Guo and Langevin, 2002; Langevin et al., 2003). Generally speaking, most variable-density models are numerically complicated and computational expensive, because of the small timestep and the implicit procedure of solving flow and transport equations iteratively many times within each timestep (Werner et al., 2013).





Karst aquifer is particularly vulnerable to groundwater contamination including

seawater intrusion in the coastal regions, since the well-developed sinkholes, karst

windows, and subsurface conduit networks are highly permeable and usually connected.

The caves are found directly open to the sea and become submarine springs below the sea

level, connected with the conduit network as natural pathways for seawater intrusion. The

highly permeable subsurface conduit network in a costal karst aquifer cause seawater

intrude significantly further landward and contaminate freshwater resources. The

preferential flow within the conduit also significantly moves the position of seawater and

freshwater mixing zone landward in karst aquifers (Calvache and Pulido-Bosch, 1997).

Fleury et al. (2007) reviewed a number of studies about freshwater discharge and

seawater intrusion through karst conduits and submarine springs in the coastal karst

aquifers, and summarized some important controlling factors, including hydraulic

gradient of equivalent freshwater head, hydraulic conductivity, and seasonal precipitation

variation. Rainfall and regional freshwater recharges significantly affect the extents of

seawater intrusion. Salinity near the outlet of conduit system is diluted by freshwater

discharge during a rainfall season, but remains constant as saline water during a low

rainfall period (Martin and Dean, 2001; Martin et al., 2012).

The Woodville Karst Plain (WKP) is a typical coastal karst aquifer, where the

Spring Creek Springs (SCS) is a first magnitude spring consisting of 14 submarine

springs located in the Gulf of Mexico (Fig. 1). SCS is also an outlet of the subsurface

conduit network, exactly located at the shoreline beneath the sea level. Tracer test studies

and cave diving investigations indicate that the conduit system starts from the submarine

spring and extends 18 km landward connecting with an inland spring called Wakulla



Spring, although the exact locations of the subsurface conduits are unknown and difficult

to explore (Kernagis et al., 2008; Kincaid and Werner, 2008). Some evidence shows that

seawater intrusion has been observed through subsurface conduit system for more than 18

km in the WKP (Xu et al., 2016). The relationship of seawater intrusion, groundwater

flow and rainfall recharges in the WKP was described by a conceptual model of

groundwater flow cycling in Davis and Verdi (2014), and then quantitatively simulated

by a CFPv2 model in Xu et al. (2015b). In addition, Davis and Verdi (2014) also point

out that sea level rise at the Gulf of Mexico in the past century could be a reason for the

increasing discharge at an inland karst spring (Wakulla Spring) and decreasing discharge

at SCS, when the hydraulic gradient towards the Gulf between the two springs decreases.

(Insert Fig. 1 here)

Modeling groundwater flow in a dual-permeability karst aquifer is challenging,

because of the non-laminar flow calculation in a karst conduit system (Davis, 1996;

Shoemaker et al., 2008; Gallegos et al., 2013). Several coupled discrete-continuum

numerical models have been developed and applied to solve the non-laminar flow in the

conduit and the Darcian flow in a porous medium simultaneously, such as MODFLOW-

CFPM1 (Shoemaker et al., 2008) and CFPv2 (Reimann et al., 2014; Reimann et al., 2013;

Xu et al., 2015a; Xu et al., 2015b). However, these models only solve constant density

governing equations, which are not applicable for simulating the density-dependent

seawater intrusion processes in a coastal aquifer. The VDFST-CFP developed by Xu and

Hu (2017) is a density-dependent discrete-continuum modeling approach to study

seawater intrusion in a coastal karst aquifer with conduits, however, is only able to

simulate synthetic cases but not yet for the field scale applications. Therefore, the



variable-density SEAWAT model is still used in this study, in which Darcy equation is

        not only used to calculate flow rate in the porous medium, but also the conduit flow rate

        with large values of hydraulic conductivity and effective porosity.

             Very few studies addressed the parameter sensitivities of seawater intrusion in a

        coastal karst aquifer. Shoemaker (2004) performed a sensitivity analysis of the SEAWAT

model for seawater intrusion in a homogeneous aquifer with porous medium, and

        concluded that dispersivity is an important parameter to the head, salinity and

        groundwater flow simulations and observations in transition zone. Shoemaker (2004) also

        concluded that salinity observations are more effective than head observation, and the

        "toe" of the transition zone is the most effective location for head and salinity simulations

and observations. The sensitivity results in this study confirms some conclusions in

        Shoemaker (2004), and highlights the significance of conduit network in simulating

        seawater intrusion in a coastal karst aquifer, mainly due to the interaction between the

        karst conduit and the porous medium.

             In this study, the local and global sensitivity analyses are conducted to evaluate

the head and salinity simulations with respect to parameters in a coupled density-

        dependent flow and transport model. To our knowledge, this is the first attempt to assess

        the parameter sensitivities for seawater intrusion in a vulnerable dual-permeability karst

        aquifer with conduit network. The rest of the paper is arranged as follows: the details of

        local and global sensitivity analysis are introduced in Sect. 2. The model setup,

hydrological conditions, model discretization, initial and boundary conditions are

        discussed in Sect. 3. The results of local and global sensitivity analysis are discussed in



Sect. 4. The scenarios of seawater intrusion simulation with different boundary

conditions and elapsed time are presented in Sect. 5. The conclusions are made in Sect. 6.

**2. Methods**

The governing equations solved in the SEAWAT model can be found in the Guo

and Langevin (2002), including the variable-density flow equation with additional

density terms, and advection dispersion solute transport equation. The local and global

sensitivity methods used in this study are briefly introduced below.


**2.1 Local sensitivity analysis**

In this study, UCODE_2005 (Poeter and Hill, 1998) is applied in the local

sensitivity analysis, which evaluates the derivatives of model simulations with respect to

parameters at the specified values (Hill and Tiedeman, 2006). The forward difference

approximation of sensitivity is calculated as the derivative of the $i$th simulation respect to

the $j$th model parameters,

$$\left.\frac{\partial y'_i}{\partial x_j}\right|_b \approx \frac{y'_i(x + \Delta x) - y'_i(x)}{\Delta x_j} \qquad 1)$$

where $y'_i$ is the value of the $i$th simulation; $x_j$ is the $j$th estimated parameter; $x$ is a vector

of the specified values of estimated parameter; $\Delta x$ is a vector of zeros except that the $j$th

parameter equals $\Delta x_j$.

The sensitivities are calculated by running the model once using the parameter

values in $x$ to obtain $y'_i(x)$, and then changing the $j$th parameter value and running the

model again in $x + \Delta x$ to obtain $y'_i(x + \Delta x)$. Scaled sensitivities are used to compare the




parameters sensitivities that may have different units. In UCODE_2005, a scaling method

is used to calculate the dimensionless scaled sensitivities (DSS) by the following equation,

$$dss_{ij} = \left(\frac{\partial y_i'}{\partial x_j}\right)\Bigg|_x |x_j| \omega_{ii}^{1/2} \qquad \text{2)}$$

where $dss_{ij}$ is the dimensionless scaled sensitivity of the $i$th simulation with respect to

the $j$th parameter; $\omega_{ii}$ is the weight of the $i$th simulation.

The DSS values of different simulations with respect to each parameter are

accumulated as the composite scaled sensitivities (CSS). The CSS of the $j$th parameter is

evaluated via:

$$css_j = \sum_{i=1}^{ND} \left[ (dss_{ij})^2 \Big|_x / ND \right]^{1/2} \qquad \text{3)}$$

where $ND$ is the number of simulated quantities, for example, the head and salinity

simulations in this study.

**2.2 Morris method in global sensitivity analysis**

The local sensitivity analysis is conceptually straightforward and easy to compute

without expensive computational cost. However, the sensitivity indices for parameters are

calculated at the specified values only, but not for the entire parameter ranges. In addition,

the indices are approximated in the first order derivative only, assuming a linear

relationship of simulated quantities with respect to parameters. The higher orders

relationship and interactions among parameters are not considered in the local sensitivity

analysis.



The global sensitivity analysis evaluates the simulations with respect to parameters within the entire parameter range, instead of the specified values in the local sensitivity analysis. Morris method is applied to evaluate the global parameter sensitivities in this study (Morris, 1991). It is a so-called "one-step-at-a-time" method (OAT), meaning that only one input parameter is perturbed and given a new value in each run. The Morris method is made by a number $r$ of local changes at different points of the possible range of input values. In each parameter, a discrete number of values called levels are chosen with the parameter ranges of variation. Two sensitivity measures are proposed by Morris method for each parameter: the mean $\mu$ that estimates the overall effect of the factor on the output, and the standard deviation $\sigma$ that estimates the ensemble of the second or higher-order effects (Saltelli et al., 2004). The mean $\mu$ and standard deviation $\sigma$ of the EEs are evaluated with the $r$ independent random paths in the Morris method,

$$\mu = \sum_{i=1}^{r} d_i / r \qquad \text{4)}$$

$$\sigma = \sqrt{\sum_{i=1}^{r} (d_i - \mu)^2 / r} \qquad \text{5)}$$

The $k$-dimensional vector $x$ of the model parameters has components $x_i$, which can be divided into $p$ uniform intervals. The effect of changing one parameter at a time is evaluated in turn by the elementary effect (*EE*), $d_i$, which is defined as,

$$d_i = \frac{1}{\tau_y} \frac{[y(x_1{}^*, \ldots, x_{i-1}{}^*, x_i{}^* + \Delta, x_{i+1}{}^*, \ldots, x_k{}^*) - y(x_1{}^*, \ldots, x_k{}^*)]}{\Delta} \qquad \text{6)}$$





where $\Delta$ is the relative distance in the coordinate; $\tau_y$ is the output scaling factor; $\{x_i{}^*\}$ is

the parameter set selected by different sampling method.

To compute the EE for the $k$ parameters, $(k+1)$ simulations are needed in the same

way as local sensitivity method for the perturbation of each parameter, which is called

one "path" (Saltelli et al., 2004). An ensemble of EEs for each parameter is generated by

multiple paths of randomly generated parameter set. The total number of calculation is

$r(k+1)$ when Monte Carlo random sampling is applied, where $r$ is the number of paths.

However, the global sensitivity analysis of this study does not apply Monte Carlo

random sampling, which needs extremely large numbers (>250) of paths to be generated

for 11 parameters. The program takes very long time to complete the random sampling

and sensitivity computation without parallelization. To save the running time and

computational cost but obtain reliable result, the trajectory sampling is developed by

Saltelli et al. (2004) and becomes a widely-used method to generate the ensembles of EEs

for Morris method in the global sensitivity analysis. The definition of $p$-level is the same

as Monte Carlo sampling, where $\Delta= \pm p/[2(p-1)]$, either positive or negative. The

trajectory method starts by randomly selecting a "base" value $x^*$ for the vector $x$. Each

component $x_i$ of $x^*$ is sampled from the set (0, $1/(p-1)$, $2/(p-1)$, … ,1). The randomly

selected vector $x^*$ is used to generate the other sampling points but not one of them,

which means that the model is never evaluated at vector $x^*$. The first sampling point, $x^{(1)}$,

is obtained by changing one or more components of $x^*$ by $\Delta$. The choice of components

$x^*$ to be increased or decreased is conditioned by that $x^{(1)}$ still being within the domain.

The second sampling point, $x^{(2)}$, is generated from $x^*$ with the property that it differs




from $x^{(1)}$ in its $i$th component, which has been either increased or decreased by $\Delta$, but

also still being within the domain. The third sampling point, $x^{(3)}$, differs from $x^{(2)}$ for

only one component $j$, for any $j \neq i$, will be $x_j^{(3)} = x_j^{(2)} + \Delta$. A succession of $(k+1)$

sampling points $x^{(1)}, x^{(2)}, ..., x^{(k+1)}$ is produced in the input parameters space called a

trajectory, with the key characteristic that two consecutive points differ in only one

component. As a result, any component $i$ of the 'base' vector $x^*$ has been selected at least

once by $\Delta$ in order to calculate one EE for each parameter.

Once a trajectory is constructed and evaluated by Morris method, an EE for each

parameter $i$, $i = 1, ..., k$, can be computed. If $x^{(l)}$ and $x^{(l+1)}$, with $l$ in the set in $(1, ..., k)$,

are two sampling points differing in their $i$th component, the EEs associated with the

parameter $i$ is,

$$d_i\big(x^{(l)}\big) = \frac{[y(x^{(l+1)}) - y(x^{(l)})]}{\Delta},$$    7)

A random sample of $r$ EEs is selected at the beginning of sampling. Each

trajectory sampling has a different starting point that is randomly generated. The points

belonging to the same trajectory are not independent, but the $r$ points sampled from each

distribution belonging to different trajectories are independent.

### 3. Model development

In this study, a two-dimensional SEAWAT model is setup to simulate seawater

intrusion through the major subsurface conduit network in the Woodville Karst Plain

(WKP) (Fig. 1). Figure 2 presents a schematic figure of the cross section in a coastal karst

aquifer with a conduit network and a submarine spring opening to the sea. The model

spatial domain is not a straight line from the submarine Spring Creek Springs to Wakulla



Spring, but a cross section along the major conduit pathway of seawater intrusion in the

coastal karst aquifer. The model spatial domain and geometry in this study is setup

exactly as the regional scale in the field, for example, the 18 km long and 91 meters deep

conduit network in the WKP.

(Insert Fig. 2 here)

However, this study only addresses the seawater intrusion through the conduit,

and the flow and solute exchanges between the conduit and the surrounding porous

medium within the cross section. In the field, seawater intrudes further landward through

the conduit network and flows into the surrounding porous medium, in both the vertical

and the horizontal direction that perpendicular to the cross section. The simulation of

saltwater flow and transport in the perpendicular direction within the porous medium is

beyond the scope of the model used in this study, which only aims to study seawater

intrusion through the conduit and the porous medium of the cross section. This

assumption of 2D model is reasonable, considering that the exchange fluxes from the

conduit to the surrounding porous medium trivially affect the seawater intrusion through

the channel, when the conduit wall exchange permeability is relatively smaller compared

with the large permeability in the conduit. The simplified model is used in this study,

since the main purpose of this study is to evaluate the role of subsurface conduit in

seawater intrusion in the coastal karst aquifer. In addition, most SEAWAT models are

setup for 2D cross section with high-resolution vertical discretization. 3D coupled

density-dependent flow and transport model is rarely seen, due to the constraint of

computational cost. The parameter sensitivities of 3D density-dependent numerical model

are even more difficult to be evaluated within a reasonable time frame, because running



the model many times with parameters perturbation are required in the sensitivities

analysis.


**3.1 Hydrological conditions**

Table 1 presents the hydrological characteristics of the Upper Floridan Aquifer

(UFA) in the WKP. The parameter values are evaluated in the following local sensitivity

analysis and applied in the seawater intrusion scenarios in Sect. 5. These parameters have

been calibrated in the regional-scale groundwater flow and solute transport models by

Davis et al. (2010), and have been applied in many previous modeling studies (Gallegos

et al., 2013; Xu et al., 2015a; Xu et al., 2015b). This study does not aim to re-calibrate the

model, since observation data in the field are insufficient, especially in the subsurface

conduit. The head and salinity measurements in the conduit are rarely available, due to

the difficulties of subsurface conduit investigation and monitoring devices installation.

(Insert Table 1 here)

The hydrological characteristics (hydraulic conductivity, specific storage and

effective porosity) of the conduit system are generally greater than the surrounding

porous medium. Hydraulic conductivity of the porous medium is assigned as 2286 m/day

(7500 ft/day), and as large as 610,000 m/day (2,000,000 ft/day) for the conduit system.

Even the hydraulic conductivity of porous medium in the study region is larger than most

alluvial aquifers, because of the numerous small fractures and relatively large pores in a

karst aquifer due to dissolution of carbonate rocks. Specific storage and effective porosity

in the porous medium are assumed as $5 \times 10^{-7}$ and 0.003, respectively. Specific storage

and effective porosity are 0.005 and 0.300 in the conduit layer, respectively. The





longitudinal dispersivity is estimated as 10 m in the porous medium, but is assumed very

small (0.3 m) in the conduit, because advection is dominated and dispersion is negligible

in the solution of transport in the conduit.

**3.2 Spatial and temporal discretization**

The grid discretization and boundary conditions of the two-dimensional

SEAWAT numerical model are shown in Fig. 3, which consists of 140 columns and 37

layers in the cross section. Guo and Langevin (2002); Werner et al. (2013) pointed out

that higher-resolution grid discretization in the vertical direction is required for modeling

the density-dependent flow. The vertical thickness of each cell is set uniformly as 3.048

m (10 ft) in this study, with significantly higher resolution than the 152 m (500 ft)

thickness in many previous constant-density modeling studies in the WKP, for example,

Davis and Katz (2007); Davis et al. (2010); Xu et al. (2015a); Gallegos et al. (2013); Xu

et al. (2015b).

(Insert Fig. 3 here)

The horizontal dimension for each cell is set uniformly 152 m (500 ft) as the

scales in the field, except columns #22 and #139, which are 15.2 m (50 ft) as the vertical

conduit network connecting the submarine spring (SCS) and inland spring (Wakulla

Spring), respectively. The sizes of spring outlets and the conduit are based on the

observations data in the field and the calibrated models in previous studies (Gallegos et

al., 2013). However, the diameter of horizontal conduit network is assumed

homogeneous in this study. The outlet of vertical conduit system is the submarine spring

(SCS) located at the shoreline at column #22. The conduit system starts from the





submarine spring, descends downward to layer #29 (nearly 100 m below sea level),

horizontally extends nearly 18 km from column #22 to column #139, and then rises

upward to the top through column #139. Seawater intrudes at the SCS on the first layer of

column #22, and then flows vertically downward into the conduit system. The inland

spring is simulated by the DRAIN package as general head boundary condition in the

SEAWAT model. All layers are simulated as confined aquifer since the conduit is fully

saturated, which are identical to the previous numerical models used in Davis et al.

(2010); Xu et al. (2015a); Xu et al. (2015b) in the WKP.

In this study, a transient 7-day stress is simulated in the SEAWAT model. The

scenarios of longer simulation time are exceptions for evaluating seawater intrusion

under an extended low rainfall period in Sect. 5.4. The timestep of flow model is set as

0.1 days, and the timestep of transport model is determined by SEAWAT automatically.

### 3.3 Initial and boundary conditions

The initial condition of head is set 0.0 m as the present-day sea level for the cells

from the left boundary to the shoreline (column #22), and gradually rises to 1.52 m (5.0

ft) at inland boundary on the right, based on the elevation of the inland spring. Please

note that values of head are written in the input files of SEAWAT model, instead of

equivalent freshwater head. The initial conditions of salinity are assumed constant along

the vertical direction in each column. Salinity at the leftmost 10 columns is set uniformly

as 35.0 PSU (Practical Salinity Unit) as seawater without mixing near the sea boundary.

The seawater/freshwater mixing zone has the salinity initial condition from 35 PSU at

column #11 to 0 PSU at column #45, with a gradient of 1.0 PSU per column. Salinity is





set uniformly as 0.0 PSU from column #46 to the inland boundary on the right, as

uncontaminated freshwater before seawater intrudes. Several testing cases have been

made to test and confirm that the initial conditions trivially affect the modeling results.

The boundary conditions are also presented in Fig. 3. The bottom of model

domain is no-flow boundary condition as the less-permeable confining unit of the UFA

base. The constant head and concentration inland boundary condition on the right is 1.5

m (5.0 ft) as the elevation of inland spring, and 0.0 PSU as uncontaminated freshwater.

The seawater boundary on the left is 3.38 km away from the shoreline, set as 0.0 m

constant head as the present-day sea level and 35.0 PSU constant concentration as

seawater without mixing. However, the boundary conditions of head and salinity at the

submarine spring (column #22, layer #1) are adjusted and evaluated in the scenarios of

different sea level, salinity and rainfall conditions in Sect. 5.

**4. Sensitivity Analysis**

Sensitivity analysis evaluates the uncertainties of salinity and head simulations

with respect to eleven parameters, provides the knowledge for understanding the

hydrological characteristics and boundary conditions in a coastal karst aquifer with a

conduit. The symbols and definitions of the eleven parameters in the SEWAT model are

listed in Table 1, as well as the specified evaluated values in local sensitivity analysis,

and the parameter ranges in the global sensitivity analysis (Table 1). There are six

parameters in the groundwater flow model, including hydraulic conductivity (HY_P and

HY_C), specific storage (SS_P and SS_C) of the conduit and the porous medium,

recharge rate (RCH) and the sea level at the submarine spring (H_SL). The other five





parameters, including effective porosity (PO_P and PO_C), dispersivity (DISP_P and

DISP_C) of the conduit and the porous medium, and the salinity at the submarine spring

(SC), are in the solute transport model.

### 4.1 Local sensitivity analysis

375        In the local sensitivity analysis, the CSS (composited scaled sensitivities) of

parameters with respect to head and salinity simulations are calculated at several

locations in the conduit and the porous medium domains. Again, the specified parameter

values in the local sensitivity analysis are consistent with the hydrological characteristics

of the UFA, and the head and salinity boundary conditions of the conduit outlet are set as

present-day sea level and seawater without mixing. The simulation results with the

parameters in the local sensitivity analysis are presented in Sect. 5.1, as the maximum

seawater intrusion case. Parameter sensitivities are computed at several locations, from

column #25 to column #75 with an interval of 5 cells along the horizontal conduit (layer

#29), where column #25 is very close to the shoreline and column #75 is basically the

uncontaminated freshwater aquifer. The parameter sensitivities of simulations in a porous

medium are evaluated at layer #24, 15.2 m (50 ft) or 5 layers above the conduit layer,

from column #25 to column #75 with an interval of 5 cells along the horizontal direction.

### 4.1.1 Local sensitivity analysis of simulations in the conduit

390        Figure 4 shows the arithmetic mean of CSS for each parameter with respect to

simulations in all evaluated locations along the conduit layer. The largest CSS value in

the local sensitivity analysis indicates that salinity at the submarine spring (SC) is the





most important parameter to both salinity and head simulations. Hydraulic conductivity, specific storage and effective porosity of the conduit (HY_C, SS_C and PO_C) as well as

sea level at the submarine spring (H_SL) are also important parameters. Hydraulic conductivity, specific storage and effective porosity of the porous medium (HY_P, SS_P and PO_P), recharge rate (RCH) and dispersivity (DISP_C and DISP_P) are the less important parameters. Generally speaking, the parameter sensitivities with respect to head simulations are similar and consistent with salinity simulations.

(Insert Fig. 4 here)

Salinity at the submarine spring (SC) is the most important parameter, because the submarine spring is the boundary condition and the major source of seawater intrusion through the conduit network in a coastal karst aquifer. Seawater enters the conduit system at the submarine spring, and continuously intrudes landward through the subsurface

conduit system. Seawater also flows into the surrounding porous medium via the exchanges between the two domains, when the equivalent freshwater head in the conduit is larger than the surrounding porous medium. Increasing salinity at the submarine spring causes further seawater intrusion through the conduit network with higher equivalent freshwater head, moves the mixing zone position further landward, and vice versa. The

salinity at the submarine spring is determined by freshwater mixing and dilution from the conduit network, in other words, is controlled by the rainfall recharges and freshwater discharge from the aquifer to the sea. In other word, freshwater dilution is represented by salinity at submarine study instead of the recharge flux on the boundary, which is unknown and difficult to quantitatively estimate in a two-dimensional numerical model

In addition, the simulations in the conduit are more sensitive to hydraulic conductivity,



specific storage and effective porosity of the conduit (HY_C, SS_C and PO_C) rather

than the hydrological characteristics of the porous medium (HY_P, SS_P and PO_P),

since conduit network is the major pathway for seawater intrusion. The sea level at the

submarine spring (H_SL) is indicated as an important boundary condition in the model,

but is not as important as the salinity at the submarine spring (SC). In other words, the

extent of seawater intrusion in the conduit is more sensitive to rainfall recharge and

freshwater discharge that represented by the parameter SC, rather than the sea level

and/or tide level variations. Recharge (RCH) is not an important parameter with respect

to simulation in this study, which only represents a small portion of total rainfall recharge

in the two-dimensional seawater intrusion model, since the rainfall recharges in the whole

springshed converge into the subsurface conduit. In this study, salinity at the submarine

spring represents the rainfall recharge with the largest CSS value, instead of the

parameter RCH.

        In the local sensitivity analysis, the conduit and porous medium dispersivities

(DISP_C and DISP_P) are not important to salinity and head simulations, because

advection is dominated in the transport of seawater within the highly permeable conduit

network, while dispersion is negligible in such rapid flow condition. This is different

from the results of parameter sensitivities in a homogeneous aquifer, where dispersivity is

actually a very important parameter. Meanwhile, the dispersion solution and dispersivity

sensitivities are hardly calculated when conduit flow becomes turbulent. On the other

hand, the numerical dispersion is significantly greater than the solution of dispersion

equation in the conduit. The Peclet number can be as great as 2500, which is obviously

beyond the Peclet number criteria (<4) for solving the advection dispersion transport





equation by finite difference method. In other words, the calculation of salinity dispersion

is inaccurate in the conduit by the SEAWAT model with large uncertainty, which brings

about the inaccurate results of dispersivity sensitivities .

Six parameters with the largest CSS are presented in Fig. 5, with respect to the

combination of head and salinity simulations in the evaluated locations along the conduit

network, from column #25 to column #75. The largest CSS values for all parameters are

found at either column #50 or #55 within the conduit, matches the position of

seawater/freshwater mixing zone along the conduit network in the maximum seawater

intrusion case in Sect. 5.1. The CSS values are relatively larger for all parameters at the

mixing zone, because head and salinity simulations only change significantly near the

mixing zone but remain constant in other locations.

(Insert Fig. 5 here)

**4.1.2 Local sensitivity analysis of simulations in the porous medium**

Figure 6 shows the arithmetic mean of CSS for each parameter with respect to

head and salinity simulations in all evaluated locations along the porous medium (layer

#24). Similar to the parameter sensitivities of simulations in the conduit, the largest CSS

indicates that salinity at the submarine spring (SC) is still the most important parameter

with respect to simulations in the porous medium, although it is a boundary condition of

the conduit system. However, some parameter sensitivities are different from the results

of simulations in the conduit. The hydraulic conductivity and effective porosity of both

the conduit and porous medium (HY_C, HY_P, PO_C & PO_P), specific storage of the

conduit (SS_C) and dispersivity of the porous medium (DISP_P) have smaller CSS than

the parameter SC, but are still identified as relatively important parameters. The other

parameters (DISP_C, RCH and SS_P) are not important to the simulations in the porous

medium. The CSS of the eight relatively important parameters are plot at different

evaluated locations along the layer of porous medium in Fig. 7. Similar to the sensitivity

analysis of simulations in the conduit, the largest CSS of each parameter is found at either

column #35 or #40, matches the mixing zone position in the porous medium in the

maximum seawater intrusion case in Sect. 5.1.

(Insert Fig. 6 and 7 here)

Salinity at the submarine spring (SC) is still the most important parameter for

simulations in the porous medium, because the submarine spring is the major source and

the entrance of seawater intrusion into the aquifer. The conduit network is the major

pathway for seawater intruding landward and flowing into the surrounding porous

medium. Similarly, the conduit hydrological characteristics, such as hydraulic

conductivity, effective porosity and specific storage (HY_C, PO_C and SS_C), are also

important parameters with respect to the simulations in the porous medium. Groundwater

flow and seawater transport through the conduit system have significant impact on the

head and salinity simulations in the surrounding porous medium. On the other hand, the

hydrological characteristics of porous medium, including hydraulic conductivity,

effective porosity and dispersivity (HY_P, PO_P and DISP_P), also have large CSS with

respect to simulations in the porous medium. It is easy to understand that head and

salinity simulations are sensitive to the in-situ hydrological characteristics of porous

medium. In summary, simulations in the porous medium are sensitive to the hydrological

characteristics of both the conduit and the porous medium, indicating that the interaction



between the two domains are important for simulating seawater intrusion in a dual-

permeability coastal karst aquifer. As a result, salinity and head observations in the

conduits and other karst features have significant meanings and values for calibrating

numerical models and for understanding seawater intrusion in a coastal karst aquifer.

**4.2 Global sensitivity analysis**

The derivatives of head and salinity simulations with respect to the selected

parameters are calculated to test if the CSS values in the local sensitivity analysis is

representative for the entire parameter range (Fig. 8). For example, head and salinity

simulations in the conduit are nearly constant to the dispersivity of the porous medium

(DISP_P), which is evaluated as an unimportant parameter in the local sensitivity study.

Simulation in the porous medium shows a linear relationship to the parameter HY_P,

with a median level CSS calculated in the local sensitivity analysis. However, both head

and salinity simulations are non-linear to salinity at the submarine spring (SC), which is

the most important parameter and the boundary condition of conduit system with the

largest CSS (Fig. 8). The local sensitivity analysis is only based on the specified

parameter value, but not representative for the entire parameter ranges and higher-order

derivatives. Therefore, the global sensitivity analysis is necessary for fully evaluating the

relationship of simulations with respect to parameters.

(Insert Fig. 8 here)

In the following global sensitivity analysis, parameter sensitivities with respect to

simulations are calculated in a specified location in the conduit (column #50, layer 29)

and the porous medium (column #35, layer #24), respectively. The parameters





sensitivities have the largest CSS in these locations, which are identical to the mixing

zone position is in the maximum seawater intrusion case in Sect. 5.1. The Trajectory

sampling method developed by Saltelli et al. (2004) is introduced in Sect. 2.2 and applied

in this study, with the recommended choice of $p = 4$ and $r = 10$ by Saltelli et al. (2004).

**5.2.1 Global sensitivity analysis to simulations in the conduit**

        In the global sensitivity analysis, the mean and standard deviation of the EEs for

each parameter with respect to salinity simulation in the conduit (column #50, layer #29)

are presented in Fig. 9a. The largest mean value indicates that parameter SC is the most

important parameter to salinity simulations. It is consistent with the result in local

sensitivity analysis, since the submarine spring is the boundary condition of the major

pathway for seawater intrusion in the numerical model. The non-linear relationship of

salinity simulation with respect to parameter SC shown in Fig.8 is the main reason for the

largest standard deviation of the EEs, since the derivatives vary at different evaluated

values. Similar to the local sensitivity analysis, the hydraulic conductivity and effective

porosity of the conduit (HY_C and PO_C), as well as sea level (H_SL), are all important

to salinity simulation with relatively large mean and standard deviation values of EEs.

These parameters are either the hydrological characteristics or the boundary conditions of

conduit network. However, other parameters are relatively less important with small

mean and standard deviation of EEs. Generally speaking, the global sensitivity study

results for salinity simulation in the conduit are similar to the local sensitivity analysis.

(Insert Fig. 9 here)





The global parameter sensitivities with respect to head simulation exhibit more

complicated (Fig. 9b). The mean and standard deviation of EEs for each parameter with

respect to head simulation are smaller than the sensitivities of salinity simulation, which

means that salinity is more sensitive than head simulation with respect to the parameters.

The mean values of EEs indicate that the specific storage (SS_C) and effective porosity

(PO_C) of the conduit are the two most important parameters with respect to the head

and salinity solutions in the coupled density-dependent flow and transport model,

respectively. The values of specific storage and effective porosity are calculated by the

percentage of void space in the aquifer, based on the physically measured data in the field.

Although salinity at the submarine spring (SC) does not have the largest mean of EEs, it

is still an important parameter for head simulation with a large standard deviation of EEs,

since the derivatives vary in different evaluated locations, with non-linear relationship to

head simulation (Fig. 8). Head simulations are also sensitive to the boundary conditions

of salinity in the transport model, because equivalent freshwater head is a function of

density in terms of salinity simultaneously in the coupled variable-density flow and

transport model.

The means and standard deviations of EEs for effective porosity and specific

storage of the conduit (PO_C and SS_C) are much larger than hydraulic conductivity

(HY_C), indicating that head simulation in the conduit is more sensitive to effective

porosity and specific storage, rather than the hydraulic conductivity. The sensitivity result

is different from the common knowledge and empirical experience in hydrogeology, but

is actually reasonable in karst modeling. The uncertainty of non-laminar flow rate

calculation in the continuum model is highlighted in the sensitivity analysis. In the





SEAWAT model, the Darcy equation estimates the specific discharge in the whole model

domain including the conduit system, however, is only accurate for laminar seepage flow

in the porous medium. Groundwater flow easily becomes non-laminar and higher than

the critical Reynolds number in the conduit system with giant diameter. In such case, the

conduit flow rate is non-linear to head gradient and beyond the applicability of Darcy

equation, which means that the Darcy equation in SEAWAT model has relatively large

error in the non-laminar conduit flow calculation. As a result, the sensitivities of

hydraulic conductivity have significant uncertainty to evaluate the permeability of

conduit system under non-laminar flow condition.

**5.2.2 Global sensitivity analysis of simulations in the porous medium**

The means and standard deviations of EEs for each parameter with respect to

salinity simulations in the porous medium are shown in Fig. 10a. The mean and standard

deviation values indicate that parameters HY_P and SC are the two most important

parameters for simulations in the porous medium. Hydraulic conductivity of the porous

medium (HY_P) is an important term for solving head and groundwater seepage velocity

in the flow equation, which then determines the advective velocity of transport equation

in the porous medium. As the boundary condition of conduit system, salinity at the

submarine spring (SC) is also important as the major source of seawater intrusion not

only in the conduit system, but also in the surrounding porous medium via exchanges

between the two domains. The global sensitivity analysis of parameter SC highlights the

significance of the interaction between the conduit and the porous medium domains in a

dual-permeability aquifer. However, salinity at the submarine spring (SC) with respect to



simulations in the porous medium has the largest CSS in the local sensitivity study, while

the CSS of hydraulic conductivity of the porous medium (HY_P) is much smaller (Fig. 6).

Parameter sensitivity of salinity at submarine spring (SC) is evaluated at the specified

value (35.0 PSU) with the largest derivative in the local sensitivity analysis (Fig. 8).

However, simulations are non-linear to the parameter SC with variable sensitivities. In

other words, global sensitivity analysis provides more comprehensive knowledge of

parameter sensitivities within the ranges. Sea level at the submarine spring (H_SL) and

effective porosity of the porous medium (PO_P) are important parameters to salinity

simulations as well. Similar to salinity at the submarine spring (SC), sea level (H_SL) is

the boundary condition of flow model in the conduit system, which has great impacts on

head and salinity solutions in the surrounding porous medium via exchange between the

two domains. The EEs of effective porosity and specific storage of the conduit (PO_C

and SS_C) also have relatively large mean and standard deviation to salinity simulation

in the porous medium, highlight the values of dynamic interaction between the conduit

and the porous medium in this study.

(Insert Fig. 10 here)

On the other hand, the mean and standard deviation of EEs of parameters indicate

that the matrix hydraulic conductivity (HY_P) is the most important parameter for head

simulation in the porous medium (Fig. 10b), as a common knowledge in groundwater

modeling. Please note this is different from the global sensitivity results of simulations in

the conduit, in which the EEs of effective porosity and specific storage of the conduit

(PO_C and SS_C) have larger mean and standard deviation values than hydraulic

conductivity (HY_C), due to the significant uncertainty of conduit flow calculation in the





continuum SEAWAT model. Although not as significant as parameter HY_C, effective

600 porosity of the porous medium (PO_P) is an important parameter with relatively large

mean and standard deviation of EEs, because equivalent freshwater head is calculated by

water density in terms of salinity in the coupled flow and transport model. On the other

hand, sea level at the submarine spring (H_SL) is also an important parameter with

respect to head simulation in the porous medium. As the boundary condition of the

605 conduit, parameter H_SL affects head solutions in the surrounding porous medium by

determining the head-dependent exchange flux between the two domains. In general, the

boundary conditions (SC and H_SL) and hydrological properties (SS_C, HY_C and

PO_C) of the conduit system have great impacts on head simulations in the porous

medium.

610   Both local and global sensitivity analysis highlight that field observations and

numerical simulations of the karst features, including the boundary conditions and

hydrological characteristics of sinkholes, karst windows and subsurface conduit system,

are important to model uncertainty analysis. Simulations in the porous medium are

sensitive to hydrological characteristics and boundary conditions of the conduit,

615 indicating the significance of conduit system on modeling seawater intrusion in a dual-

permeability karst aquifer. The conduit system serves as the major pathway for seawater

intrusion further landward and groundwater contamination in the aquifer. On the other

hand, dispersivity is no longer an important parameter in this study, compared with the

sensitivity analysis in Shoemaker (2004) that dispersivity is found as an important

620 parameter, in which a homogeneous porous medium domain is evaluated without the

preferential advective flow. In a dual-permeability karst aquifer system, advection

transport is dominated in the conduit and the surrounding porous medium as well, while dispersion becomes relatively less important. In this study, the uncertainty of dispersiviy sensitivities can be significant, since the large Peclet number in the conduit is beyond its

criteria for solving transport equation by finite difference method. An experiment of inactivating DSP package in SEAWAT confirms that the mixing is mostly due to the numerical dispersion instead of the solution of dispersion equation in this study.

    In general, parameter sensitivities of simulations in the porous medium are more complicated than those in the conduit, especially for the head simulations. The global

sensitivity analysis provides an insight of the parameter interactions and the higher-order relationship with respect to simulations.

## 5. Seawater Intrusions in Scenarios

    Sensitivity analysis finds that salinity at the submarine spring (SC) is the most

important parameter with respect to head and salinity simulations in both the conduit and the porous medium. Sea level at the submarine spring (H_SL) is the other important boundary condition of the conduit system in the SEAWAT model. The hydrological conditions of the aquifer are represented by these two boundary conditions, such as rainfall and regional freshwater recharges (SC), and the sea/tide level at the submarine

spring (H_SL). In order to evaluate the impacts of boundary conditions on seawater intrusion in a dual-permeability system, the extents of seawater intrusion under scenarios of boundary conditions are simulated by the SEAWAT model and quantitatively measured. In addition, the length of elapsed time in simulation is constant in the sensitivity analysis for consistent comparison purposes. The extents of seawater intrusion




in a coastal karst aquifer during an extended low rainfall periods are evaluated by

extending the elapsed time in simulation in Sect. 5.4. In each scenario, only one

parameter is adjusted and others remain the same as the original values in the maximum

seawater intrusion case.

**5.1 The maximum seawater intrusion case**

        The head and salinity boundary conditions at the submarine spring are set as 0.0

m as the present-day sea level, and 35.0 PSU as seawater without mixing filled in the

conduit system outlet, respectively. In the local sensitivity analysis, parameter

sensitivities are evaluated at the specified parameter values in this case. This case is

called the maximum seawater intrusion case, in which the longest distance of seawater

intrusion is simulated by assuming that freshwater dilution by rainfall recharge is

negligible in the entire aquifer, and the outlet of conduit system is filled with seawater

without mixing. Figure 11 presents the salinity and head simulations in the cross section

with a 7-day simulation. This case is also set as the benchmark for the following

scenarios.

 (Insert Fig. 11 here)

        According to the Ghyben-Herzberg relationship, high-density seawater intrudes

landward through the deep aquifer beneath the freshwater flowing seaward on the top.

The equivalent freshwater head at the submarine spring is calculated as 2.29 m (7.5 ft)

when salinity is 35.0 PSU at the submarine spring, and seawater without mixing is filled

with the 91 meters deep submarine cave of the conduit. The equivalent freshwater head is

higher than the 1.52 m (5.0 ft) constant head at the inland spring, diverts the hydraulic



gradient landward and causes seawater to intrude into the aquifer. Seawater moves

significantly further landward through the highly permeable conduit network, also

gradually intrudes landward in the surrounding porous medium via exchange on the

conduit wall. The position of seawater/freshwater mixing zone in the deep porous

medium beneath the conduit is only slightly behind the seawater front in the conduit,

because high-density saline water easily moves downward from the conduit into the

deeper aquifer. The area with relatively smaller salinity to the left of the vertical conduit

network near the shoreline is due to the freshwater discharge dilution from the aquifer to

the sea, since the equivalent freshwater head only increases at the submarine spring but

remains constant as 0 m in other areas. Simulation result shows that the mixing zone

position in the conduit, defined as the location with salinity of 5.0 PSU, reaches nearly

5.80 km landward from the shoreline. The width of mixing interface, defined as the

distance between the locations with salinity of 1.0 PSU and 25.0 PSU, is about 7 grid

cells or 1.13 km (0.7 miles) in both the conduit and porous medium.

**5.2 Salinity variation at the submarine spring (SC)**

Sensitivity analysis indicates that the salinity at the submarine spring (SC) is

generally the most important parameter with respect to simulations in both the conduit

and the porous medium. In this study, rainfall and regional freshwater recharges are

simulated by salinity at the submarine spring instead of time-variable flux on the

boundary condition Salinity at the submarine spring is diluted by large amount of rainfall

recharge and freshwater discharge after a significant precipitation event, but remains high

after an extended low rainfall period as the maximum seawater intrusion case in Sect. 5.1.


The equivalent freshwater head at the submarine spring is calculated as 2.29 m (7.5 ft) when seawater without mixing is filled with the conduit system, however, proportionally decreases with the salinity at the submarine spring to 0.0 m, when salinity is 0.0 PSU and freshwater is filled with the conduit system. The impact of freshwater recharge on

seawater intrusion is evaluated in four scenarios with different salinity at the submarine spring of 0.0 PSU, 10.0 PSU, 20.0 PSU and 30.0 PSU. The head boundary condition is kept constant as 0.0 m as present-day sea level (Fig. 12). The mixing zone positions in both the conduit and porous medium are located at 4.0 (4.5) km away from the shoreline in the cases of salinity of 10.0 (20.0) PSU at the submarine spring. Rainfall recharge and

freshwater discharge move the interface significantly seaward. In addition, the mixing zone is very close to the shoreline in the case of salinity of 0.0 PSU at the submarine spring, when seawater intrusion is suspended and blocked by large amount of freshwater dilution. The shape of mixing interface is similar to the maximum seawater intrusion benchmark. However, the width of mixing interface is much wider due to the smaller or

even reversed hydraulic gradient from the aquifer to the sea. In such scenarios, the solution of dispersion equation is more important in salinity simulation with slower groundwater seepage flow. Generally speaking, seawater no longer intrudes significantly inland after a heavy rainfall event, and the mixing interface moves seaward when freshwater dilutes the salinity at the submarine spring and the conduit network.

(Insert Fig. 12 here)



### 5.3 Sea level variation at the submarine spring (H_SL)

Sensitivity analysis indicates that sea level at the submarine spring is also an important parameter Approximate 1.0 m sea level rise at the beginning of next century is

predicted by IPCC (2007), with significant impacts on seawater intrusion in a coastal karst aquifer. The extents of seawater intrusion in the conduit and porous medium under 0.91 m (3.0 ft) and 1.82 m (6.0 ft) sea level rise conditions are quantitatively evaluated in this study (Fig. 13). Salinity at the submarine spring remains 35.0 PSU as same as the maximum seawater intrusion benchmark, but the head at the submarine spring increases

to 0.91 m (3.0 ft) and 1.82 m (6.0 ft) as rising sea level, respectively. The width and shape of the mixing zone are similar to the simulation result in the maximum seawater intrusion benchmark. However, the mixing zone position moves landward in the conduit at almost 7.08 km from the shoreline when sea level rises 0.91 m (3.0 ft), which is 1.28 km further inland than the simulation with the present-day sea level. In the other extreme

case of 1.82 m (6.0 ft) sea level rise, seawater intrudes additional 0.97 km further inland along the conduit than the case with 0.91 m (3.0 ft) sea level rise, or 2.25 km further inland than the simulation with present-day sea level. Compared with a homogeneous aquifer, seawater intrudes further landward through the conduit network in such a dual-permeability aquifer. The modeling of sea level rise confirm the concerns of severe

seawater intrusion in the coastal region, also highlight the impacts on a karst aquifer with conduit system as the major pathway for seawater intrusion. In addition, sea level rise influence the regional flow field and hydrological conditions in a coastal aquifer. In Davis and Verdi (2014), increasing groundwater discharge at the inland Wakulla Spring in the WKP was observed associated with sea level rise in the past decades. The





relationship between spring discharge and sea level was quantitatively simulated by a

CFPv2 numerical model in Xu et al. (2015b). However, the changes of flow field and

hydrological conditions are not addressed and simulated in this study

(Insert Fig. 13 here)

**5.4 Extended low rainfall period**

        The extents of seawater intrusion under scenarios of extended low rainfall periods

are presented in Fig. 14, although the elapsed time in simulation is not evaluated in the

sensitivity analysis. The simulated time period is extended to 14, 21 and 28 days, with the

conditions of salinity and sea level at the submarine spring as 35.0 PSU and 0.0 m,

respectively, as same as the maximum seawater intrusion benchmark.

(Insert Fig. 14 here)

        During an extended low rainfall period, seawater intrudes through both the

conduit and the porous medium domains with time, since the 2.29 m (7.5 ft) equivalent

freshwater head at the submarine spring is higher than the inland freshwater boundary.

The mixing zone position also keeps moving landward slowly and persistently.

Compared with the maximum seawater intrusion benchmark with a stress period of 7-day

elapsed time in simulation, the mixing zone position after the 14-day simulation moves

additional 1.29 km landward in the conduit and the surrounding porous medium. In the

prediction of SEAWAT model, the mixing zone finally moves to 7.56 (7.89) km from the

shoreline after the 21 (28)-day extended low rainfall period. Above all, seawater keeps

intruding further inland through conduit network during an extended low rainfall period.



In such condition, the contamination of fresh groundwater resources in the aquifer

becomes an environmental issue in coastal regions.

**6. Summary and Conclusion**

        In this study, a 2D SEAWAT model is developed to evaluate the parameter

sensitivities and quantitatively estimate seawater intrusion in a dual-permeability coastal

karst aquifer with a conduit network. In the local sensitivities analysis, the composite

scaled sensitivities (CSS) are calculated to assess the salinity and head simulations in the

conduit and the porous medium at specified parameter values. Salinity at the submarine

spring (SC) is identified as the most important parameter to the simulations in both two

domains, because the submarine spring is the major entrance of seawater intrusion into

the conduit as pathway in the aquifer. The boundary conditions and hydrological

characteristics of the conduit, including sea level at the submarine spring (H_SL),

hydraulic conductivity (HY_C) and effective porosity (PO_C) are important to the

simulations in the conduit as well. On the other hand, the simulations in the porous

medium also are sensitive to the boundary conditions (SC and H_SL) and hydrological

characteristics of the conduit, such as specific storage and effective porosity (SS_C and

PO_C), due to the interaction and exchange between the two domains. Sensitivity

analysis indicates that the observations and simulations in the conduit are especially

important for understanding hydrogeological processes and modeling seawater intrusion

in such a dual-permeability karst aquifer. In addition, the largest CSS values of parameter

sensitivities can be found around the mixing zone position. The local sensitivity analysis

in this study confirms the conclusions of sensitivity studies in a homogeneous aquifer in





Shoemaker (2004), also highlights the values of conduit network in modeling seawater

intrusion in a coastal karst aquifer.

The global sensitivity analysis indicates that head simulation in the conduit is

more sensitive to effective porosity (PO_C) and specific storage of the conduit (SS_C),

instead of hydraulic conductivity. The conduit flow easily becomes non-laminar and

beyond the capability of Darcy equation in SEAWAT model, which assumes a linear

relationship between specific discharge and head gradient. Therefore, the uncertainty of

conduit permeability is difficult to be accurately evaluated by hydraulic conductivity in

the continuum model. Different from the local sensitivity study, hydraulic conductivity of

the porous medium (HY_P) has the largest mean of EEs with respect to head

simulation in the porous medium Simulations are non-linear to parameter SC, which has

the largest derivative in the specified evaluated location in the local sensitivity study.,

Dispersivity is no longer an important parameter for simulations in the conduit, which is

different from Shoemaker (2004), because advection is dominated in the solution of

saline water transport with turbulent flow in the conduit, as well as the relatively fast

seepage flow in the surrounding porous medium. In the salinity profile, the mixing is

mostly due to numerical dispersion instead of the solution of dispersion equation, since

Peclet number is extremely large and beyond the criteria of solving transport equation by

finite difference method.

Seawater intrusion is quantitatively estimated with variations of salinity and sea

level at the submarine spring, which are identified as important parameters in sensitivity

study. Seawater intrudes significantly further landward through the conduit, and flows

into the surrounding porous medium via the exchange on the pipe wall. Salinity and head

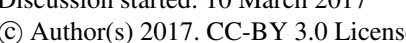


boundary conditions of 35.0 PSU and 0.0 m, respectively, are set at the submarine spring

in the case of the maximum seawater intrusion. The mixing zone position in the conduit

moves to 5.80 km from the shoreline with 1.13 km wide after a 7-day low rainfall period.

Rainfall and regional recharges dilute the salinity at the submarine spring (SC), and

significantly shift the mixing zone position seaward to 4.0 (4.5) km away from the

shoreline with salinity of 10.0 (20.0) PSU. Compared with the benchmark, seawater

intrudes additional 1.29 (2.25) km further landward along the conduit under 0.91 (1.82)

m sea level rise at the submarine spring (H_SL). In addition, the impacts of an extended

low rainfall period on seawater intrusion through conduit network are also quantitatively

assessed with longer elapsed time in simulation. The mixing zone position moves to 7.56

(7.89) km from the shoreline, after a 21 (28)-day low precipitation period.

        In a summary, the conduit network is important as the major pathway for seawater

intrusion in the aquifer, and the submarine spring is the major entrance of seawater

intrusion into the conduit. Some parameters, such as dispersivity and hydraulic

conductivity in the conduit, have different sensitivities from the previous studies for a

homogeneous aquifer.  More attentions on the modeling and field observations in the

karst features, including the subsurface conduit network, the submarine spring and karst

windows, are meaningful and important for calibrating the model and for understanding

seawater intrusion in a coastal karst aquifer.

**Competing interests**

        The authors declare that they have no conflict of interest.






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





Table 1. The symbols and definitions of parameters used in this study, the specified evaluated values in local sensitivity study and evaluation ranges (the lower and upper constraints) of each parameter in global sensitivity analysis.

| Parameter | Definitions | Lower | Upper | Evaluated value | Unit |
|---|---|---|---|---|---|
| HY_P | Hydraulic conductivity (porous medium) | 1.524 | 4.572 | 2.286 | meter/day ($\times 10^3$) |
| HY_C | Hydraulic conductivity (conduit) | 3.048 | 9.144 | 6.096 | meter/day ($\times 10^5$) |
| SS_P | Specific storage (porous medium) | 4.00 | 6.00 | 5.00 | dimensionless ($\times 10^{-7}$) |
| SS_C | Specific storage (conduit) | 0.03 | 0.07 | 0.05 | dimensionless |
| RCH | Recharge rate on the surface | 0.00 | 0.03 | 0.01 | meter/day |
| H_SL | Sea level at the submarine spring | -0.305 | 0.914 | 0.305 | meter |
| PO_P | Porosity (porous medium) | 0.001 | 0.005 | 0.003 | dimensionless |
| PO_C | Porosity (conduit) | 0.200 | 0.400 | 0.300 | dimensionless |
| SC | Salinity at the submarine spring | 0.0 | 35.0 | 35.0 | PSU |
| DISP_P | Longitudinal dispersivity (porous medium) | 6.10 | 12.20 | 10.00 | meter |
| DISP_C | Longitudinal dispersivity (conduit) | 0.15 | 0.60 | 0.30 | meter |





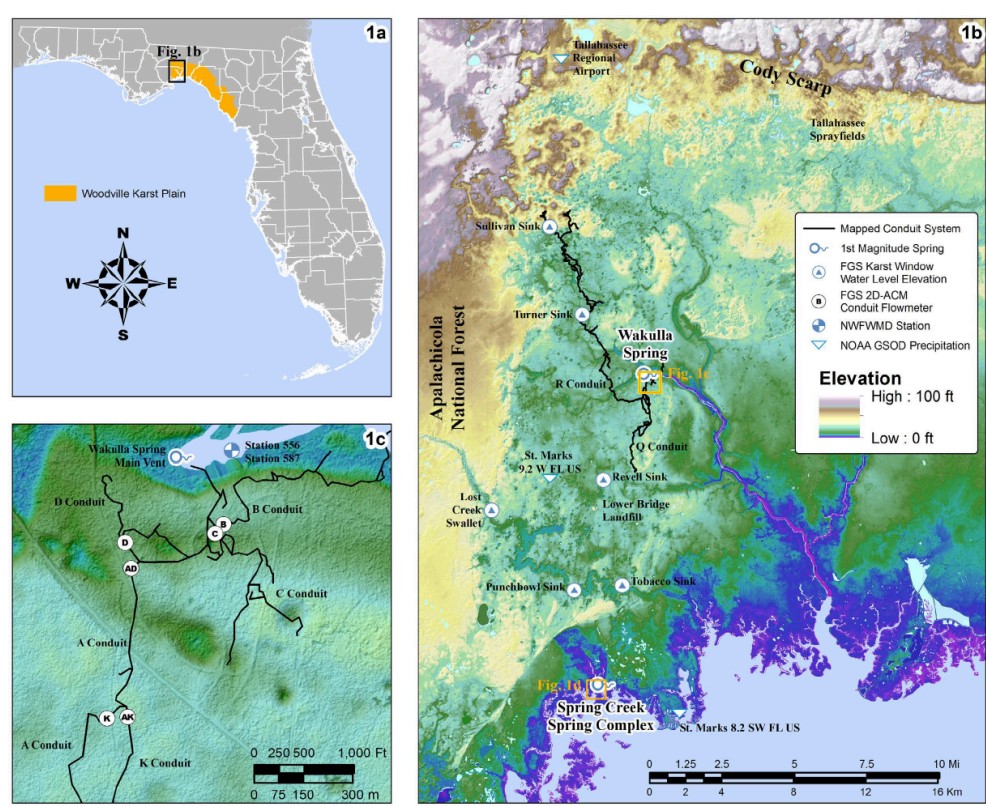

Figure 1. a) Locations of the Woodville Karst Plain (WKP) and the study site; b) The
map of the Woodville Karst Plain showing the locations of features of note with the study;
c) The detail of cave system near Wakulla Springs. Modified from Xu et al., (2016).





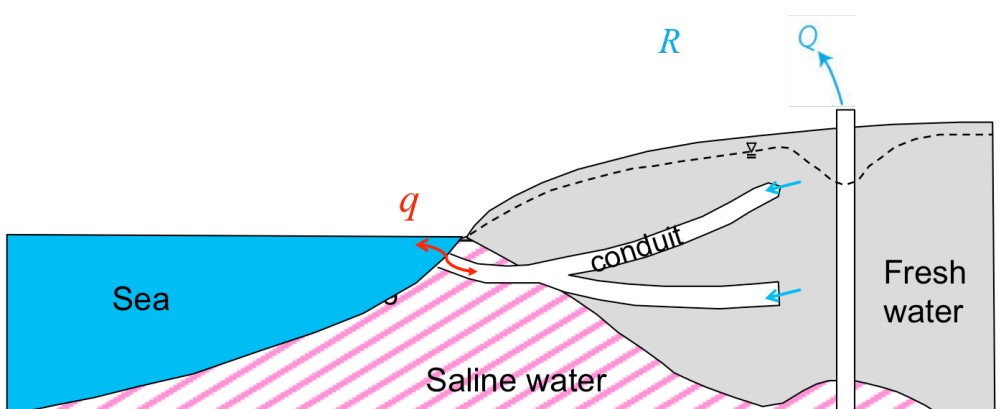

Figure 2. Schematic figure of a coastal karst aquifer with conduit networks and a submarine spring opening to the sea in a cross section. Flow direction $q$ would be seaward when sea level drops, pumping rate $Q$ is low and precipitation recharge $R$ is large; however, reversal flow occurs when sea level rises, pumping rate $Q$ is high or precipitation recharge $R$ is small.





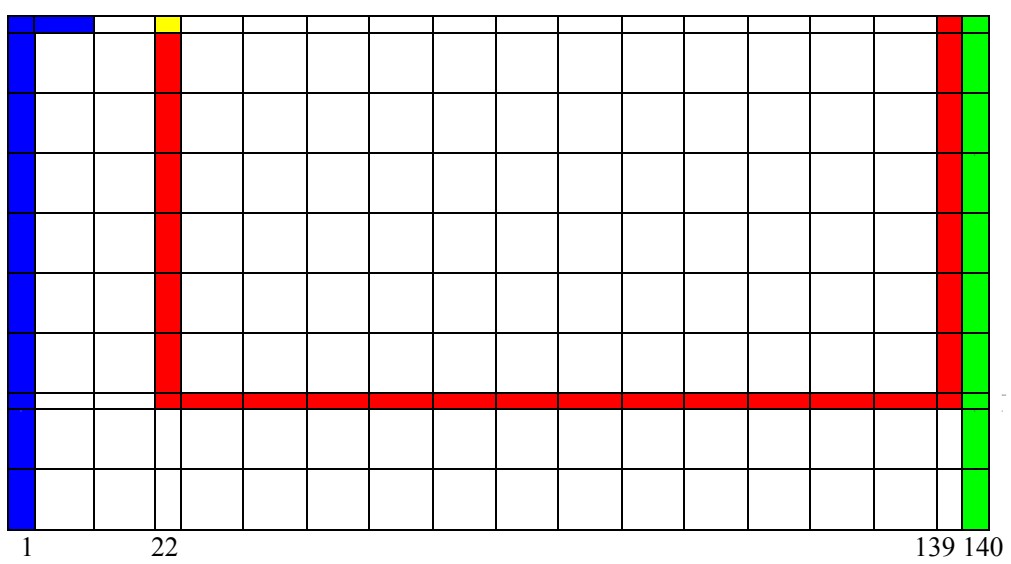

1          22                                                    139 140

Explanations:

Constant head and constant concentration of the submarine spring and outlet of karst conduit system, however, various in different cases of numerical models

Sea-edge boundary: constant head (0.0 ft in normal sea level case) and constant concentration (35 PSU)

Inland boundary: constant head (5.0 ft) and constant concentration (0 PSU)

Conduit: high hydraulic conductivity, porosity and specific storage

Porous medium: low hydraulic conductivity, porosity and specific storage

Figure 3. Schematic figure of finite difference grid discretization and boundary conditions applied in the SEAWAT model. Every cell represents 10 horizontal cells and 4 vertical cells, except the boundary and conduit layer in color with smaller width. The submarine spring is located at column #22, layer #1, and the inland spring is located at column #139, layer #1. The conduit system starts from the top of column #22, descends downward to layer #29, horizontally extends to column #139, and then rises upward to the top through column #139.



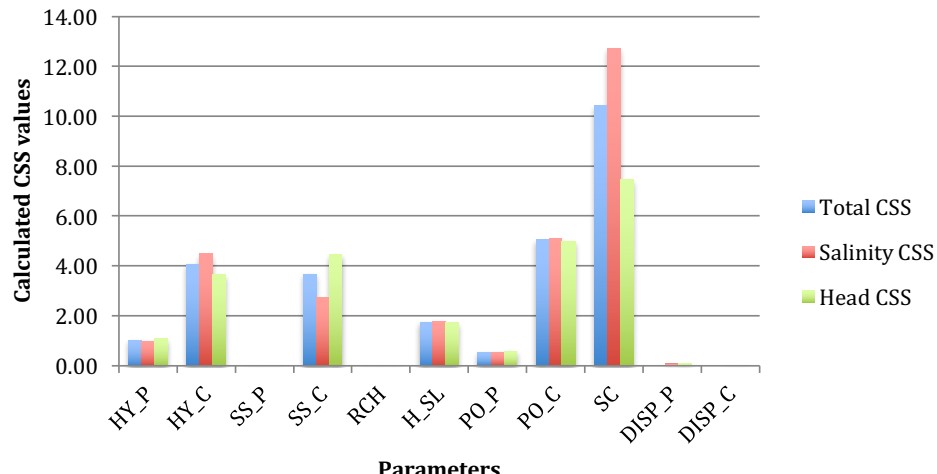

Figure 4. The CSSs (Composite Scaled Sensitivities) of all parameters with respect to simulations in the conduit (layer #29) in the local sensitivity analysis.

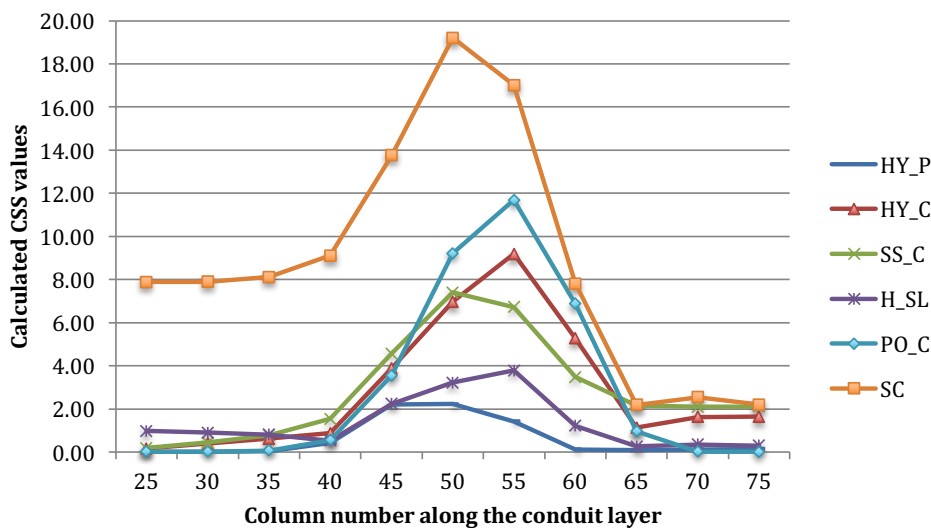

Figure 5. The CSSs (Composite Scaled Sensitivities) of selected parameters at different locations along the conduit layer (from column #25 to column #75) in the local sensitivity analysis.




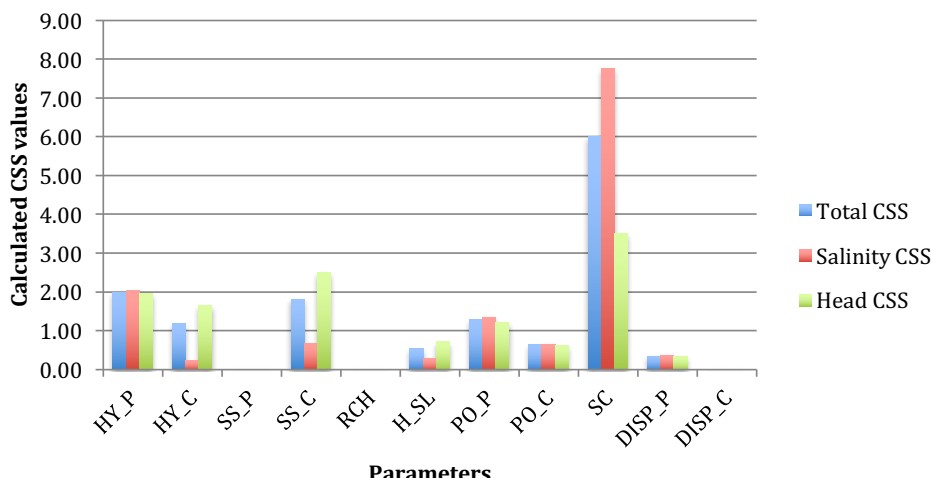

Figure 6. The CSSs (Composite Scaled Sensitivities) of all parameters with respect to simulations in the porous medium (layer #24) in the local sensitivity analysis.

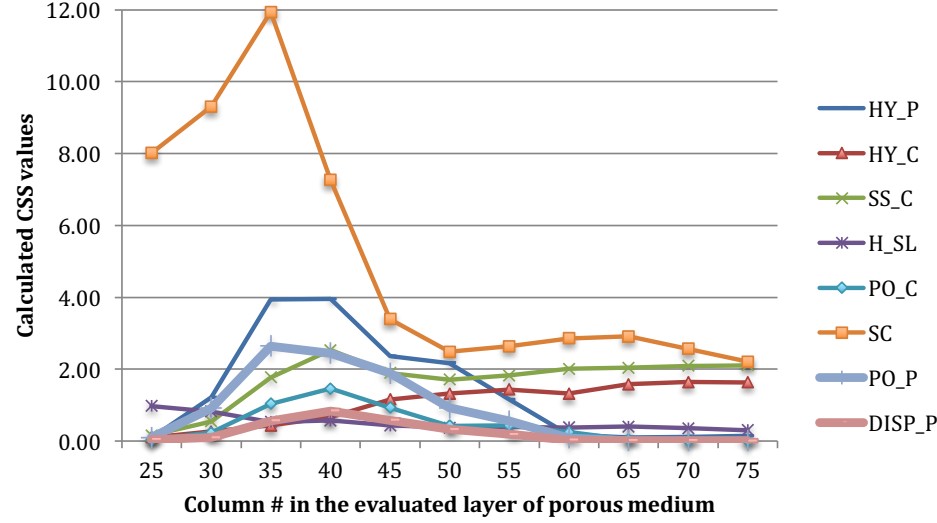

Figure 7. The CSSs (Composite Scaled Sensitivities) at different locations in the porous medium (from column #25 to column #75 at layer # 24) in the local sensitivity analysis.





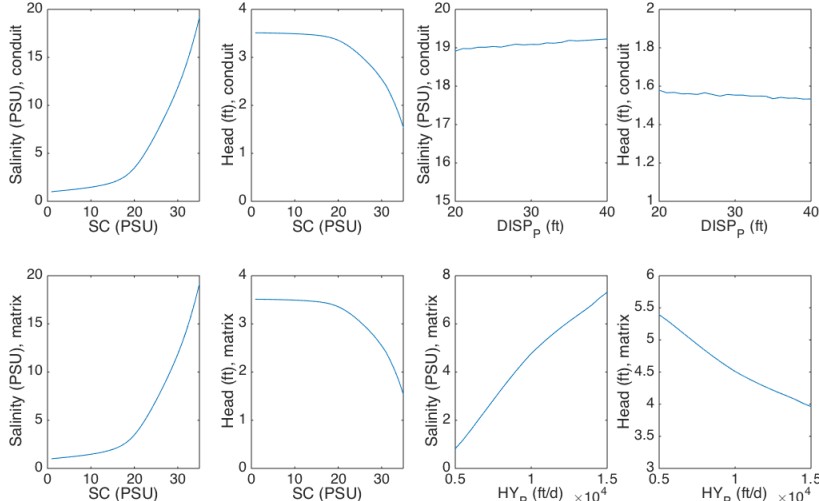

Figure 8. The non-linear relationship between head and salinity simulations with respect to parameters SC, DISP_P and HY_P. (Note that the scale for each plot is different).





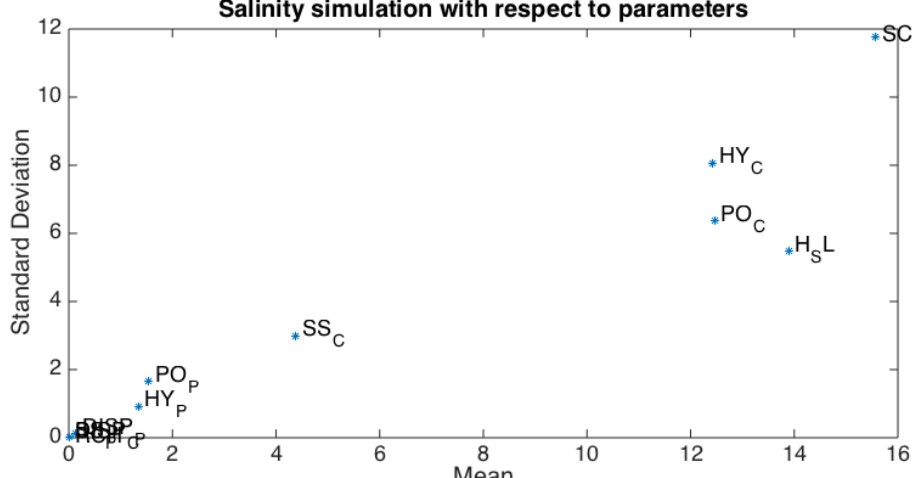

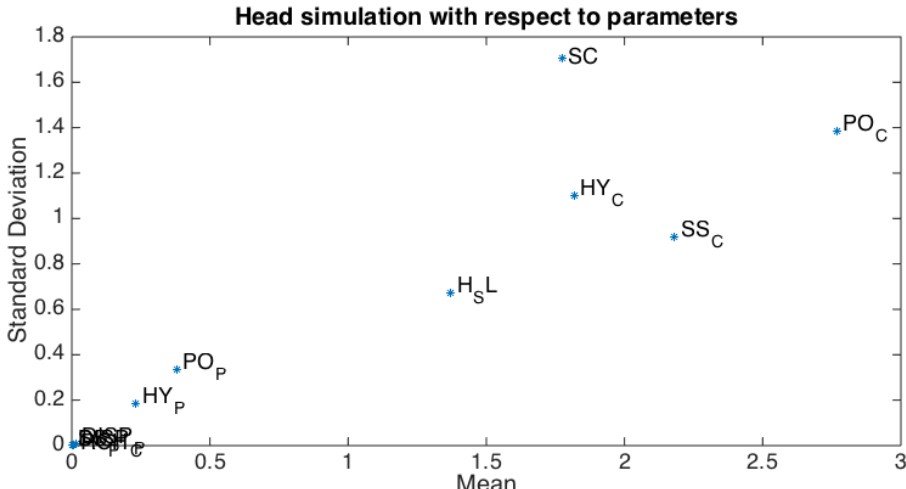

Figure 9. Mean and standard deviation of the EEs (elementary effects) of parameters with respect to simulations in the conduit (column #50, layer #29) by the trajectory sampling Morris method in the global sensitivity analysis: a) salinity simulation (top); b) head simulation (bottom).





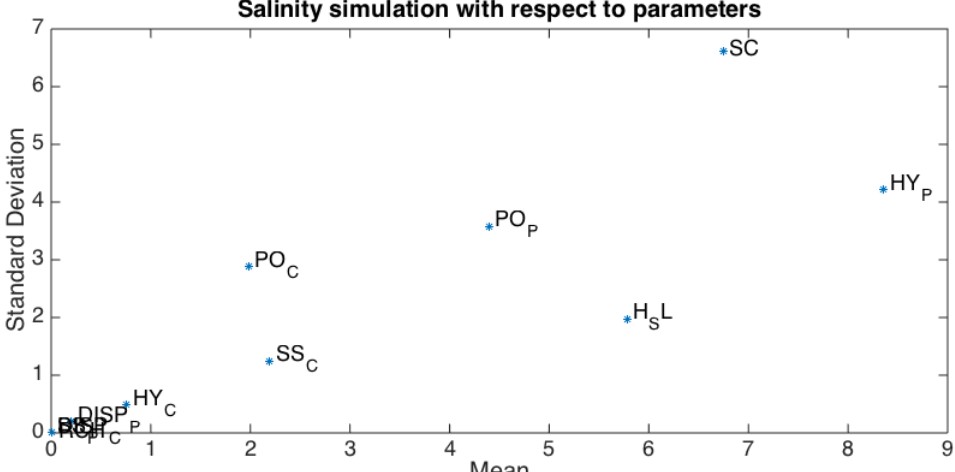

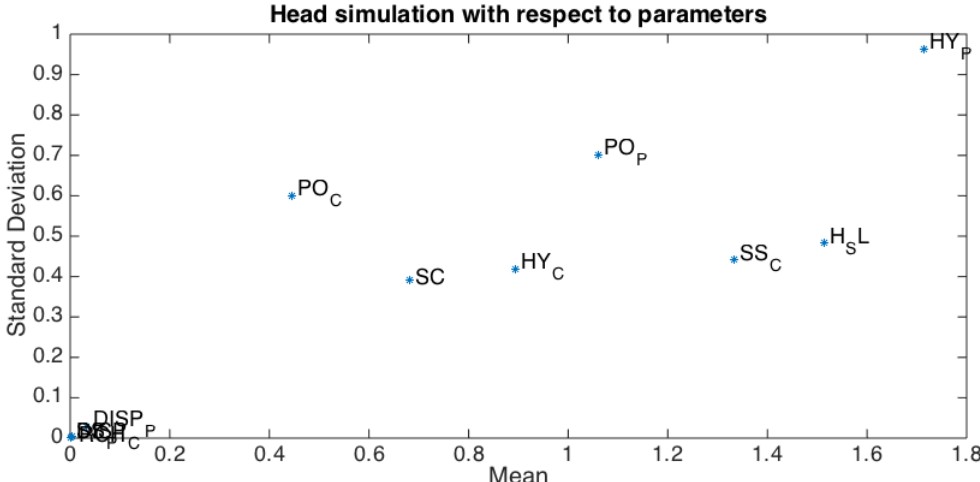

Figure 10. Mean and standard deviation of the EEs (elementary effects) of parameters with respect to simulations in the porous medium (column #35, layer #24) by trajectory sampling Morris method in the global sensitivity analysis: a) salinity simulation (top); b) head simulation (bottom).




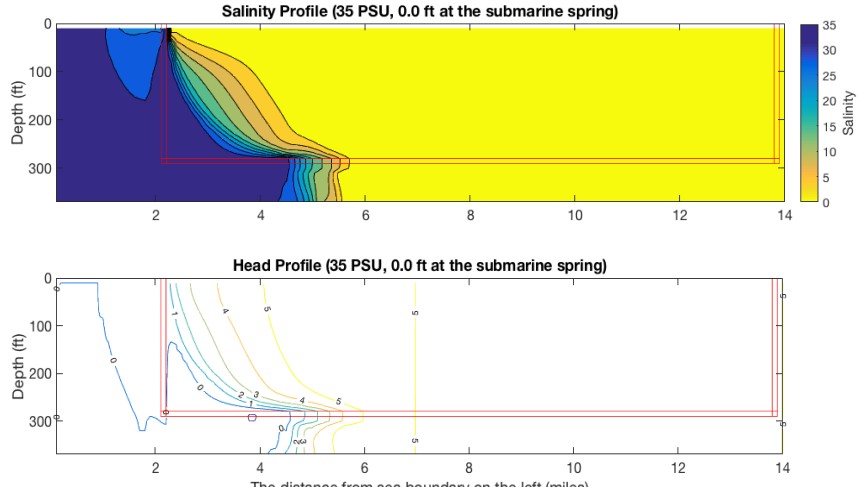

Figure 11. Salinity (top) and head (bottom) simulations of the maximum seawater intrusion case (35 PSU, 0.0 ft at the submarine spring).





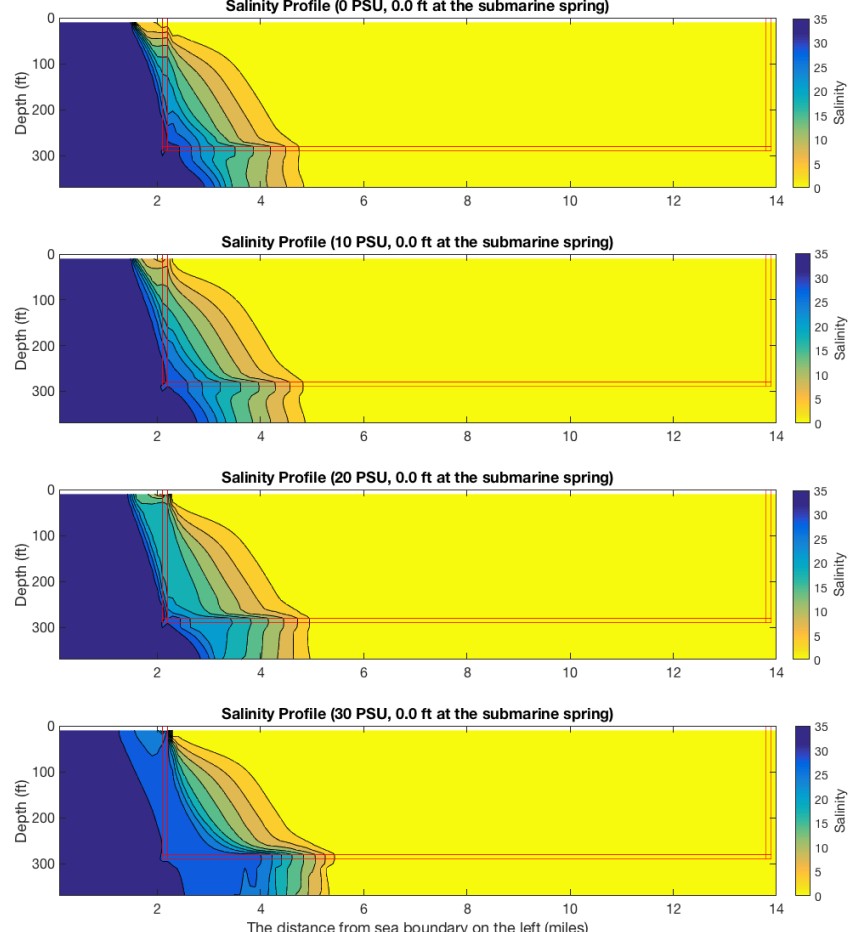

Figure 12. Salinity simulation of seawater intrusion with various salinity at the submarine spring, indicating different rainfall recharge and freshwater discharge conditions: A) 0.0 PSU, 0.0 ft at the submarine spring; B) 10.0 PSU, 0.0 ft at the submarine spring; C) 20.0 PSU, 0.0 ft at the submarine spring; D) 30.0 PSU, 0.0 ft at the submarine spring (from top to bottom).




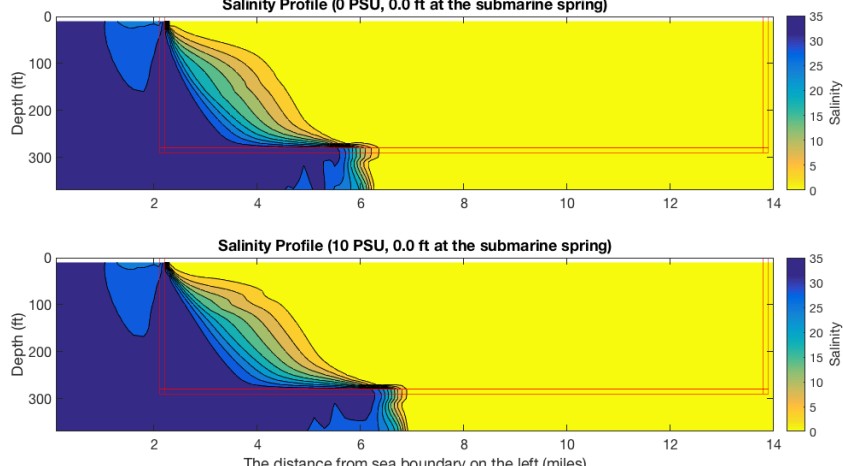

Figure 13. Salinity simulation of seawater intrusion with various sea level conditions: A) 35.0 PSU, 3.0 ft at the submarine spring; B) 35.0 PSU, 6.0 ft at the submarine spring (from top to bottom).





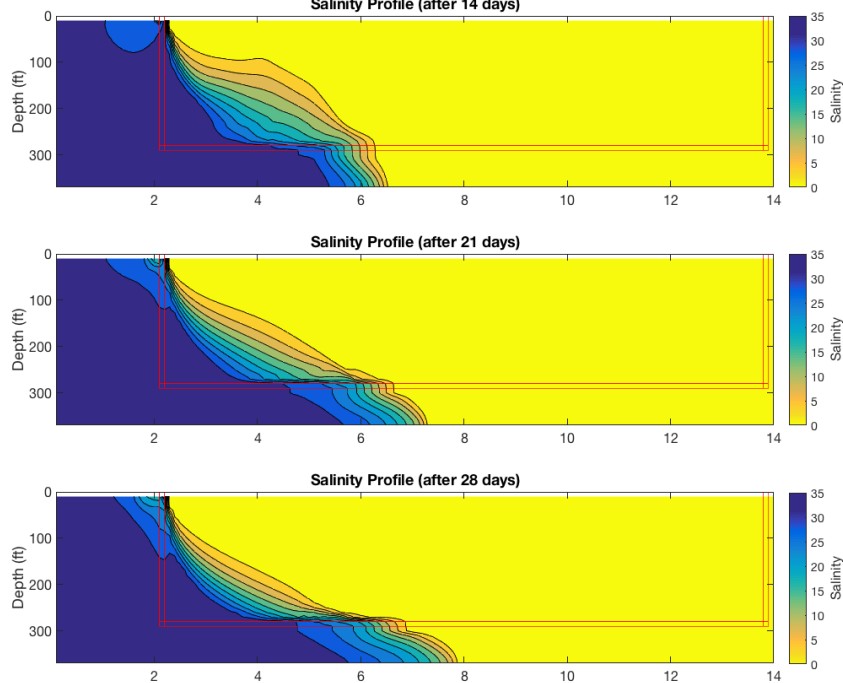

Figure 14. Salinity simulation of the maximum seawater intrusion case (35 PSU, 0.0 ft at the submarine spring) with extend simulation time during a low rainfall period: A) 14-day simulation period; B) 21-day simulation period; C) 28-day simulation period (from top to bottom).