# Peer review of "Numerical modeling and sensitivity analysis of seawater intrusion in a dual-permeability coastal karst aquifer with 5 conduit networks"

_Hydrology and Earth System Sciences, 2017_

## Referee Comment (RC1) · Anonymous Referee #1 · 19 Apr 2017

In the submitted paper Xu et al. apply local and global sensitivity analysis on a density driven distributed model (SEAWAT) to simulate a coastal aquifer I Florida (US). They use the knowledge of previous studies to define the boundary conditions and initial parameter sets of the model. Then they apply a local sensitivity analysis on the 11 model parameters in respect to various output variables of the simulated matrix and conduit systems. The same analysis is repeated with a global sensitivity analysis method (Morris) to account for interactions among the model parameters. The parameter describing the salinity at the submarine spring outlet was found to be most sensitive but also the parameters describing the conduit properties were found to influence both the conduit and the matrix behaviour. The results of the more elaborate global sensitivity analysis

scheme differ for several model parameters indicating that parameter interactions have to be considered. Finally, the authors use the simulations obtained by the sensitivity analysis to conclude about the sensitivity of the karst system to external changes and about the most gaining observations concerning model parameter identification.

The paper tackles a very interesting field of research, which is the evaluation of distributed models via sensitivity analysis. The authors clearly show that such analysis provides valuable understanding of the model and system and they also highlight that the choice of the sensitivity analysis method has strong impact on the results and conclusions. For those reasons I definitely recommend this paper for publication in Hydrology and Earth System Sciences. However, some weaknesses have to be removed first:

1. The paper is much too long. In their last paragraph of the conclusions (and also in the abstract) the authors clearly state the main outcomes of their research. However, within the body of the manuscript, they lose themselves in details to often.

2. The usage of two sensitivity analysis schemes provides a lot of insights into their differences. However, the authors do not explain why they actually compare them. For many modellers the interaction among parameters is an accepted fact. So for the sake of focus and length of the manuscript: Is it really necessary including the local sensitivity analysis? If not, delete. If yes, provide more explanation why.

3. Some more links between the model setup and field observations/previous work is necessary. It is clear that a lot of previous work was done at the study site. But sometimes it would be helpful providing some summarizing information in addition to the reference to the previous studies.

4. The elaboration of Morris's method has to be improved.

5. A clear discussion relating these results to the result of other is missing.

6. No state of the art of sensitivity analysis is missing (and no comparison to other

sensitivity analysis studies with lumped of distributed approaches in karst).

I think these corrections can all be done within the frame of moderate revisions. Please find some more specific comments in the attached and commented pdf.

Please also note the supplement to this comment:
http://www.hydrol-earth-syst-sci-discuss.net/hess-2017-85/hess-2017-85-RC1-supplement.pdf

**Supplement:**

[revised manuscript text omitted]

---

## Referee Comment (RC2) · Anonymous Referee #2 · 11 May 2017

The paper by Xu et al. presents the analysis of results with using local and global
sensitivity analysis and different scenarios on a karst coastal aquifer system with sea-
water intrusion. The topic is potentially very interesting but to be useful to the reader,
the paper needs significant improvements. The main concern is about the overall goal
of the study. The abstract concentrate mainly on the comparison between local and
global sensitivity results, but then a significant portion of the paper deals with scenar-
ios. From a first reading, it is not clear the purpose of developing all these scenarios
and how this connects to the previous sensitivity analysis. From my understanding, the
scenarios are developed based on the most sensitive parameters and are expected
to show how changes in those parameters can affect the results, but this needs to be

more clearly explained. I have serious concerns about how the local sensitivity results are presented: the analysis lacks completely the presentation of the parameter correlation coefficients which are provided as output by UCODE, but are not presented here. CSS without parameter correlation coefficients is not informative and needs to be combined to the correlation analysis. Furthermore, there is not good explanation of which observations are used in this analysis. There should be heads and salinity field observations, but it is not clear how many, where and mostly which is the weight which is applied to each observation. How are the local sensitivity analysis results used to build the simulations for global sensitivity analysis? Are the same parameters included in both the analyses? Which are the key additional information that we are getting from this double analysis? Often local sensitivity results can be used to discriminate which parameters should be included in the global sensitivity analysis, but this does not seem to be the case in this paper. The two analyses seem to be performed exactly on the same set of parameters. The introduction is too long and does not get exactly to the point of the paper. Sensitivity analysis and calibration of similar models have already been performed and it would be good to point out where. The discussion of the sensitivity analysis results and of the different scenarios should be organized and made more concise. There are also some inconsistencies with the parameter names that should be checked. My suggestion would be to develop a summary table with all the results for both sensitivity and again for the scenarios to help the reader understanding which are the key findings of this analysis. Some more specific comments include: l. 48-49: are the parameters mentioned here calibrated parameters?

l. 132: Which are the issues with VDFST-CFP? Why cannot be used in this case? Without explanation it is not clear why this other method should be mentioned.

L. 207: please spell out EEs

l. 287: the goal here seems to be just performing sensitivity analysis and not calibration. But for running the scenarios, it should be clarified how the values of the parameters have been selected. After which type of calibration? Or the values calibrated in

previous studies have been used?

l. 453: why the arithmetic mean of CSS?

l.470: this was already said at line 455. The discussion should be more compact and better organized.

l. 568: "groundwater seepage velocity" : are these used as observations?

l. 610: clarify which are the field observations and how they have been weighted

l. 619-621: check the English

l. 626: the DSP package was not mentioned earlier

l. 630: local sensitivity allows as well to understand interactions and correlations between parameters but it is not presented here

Chapter 5: how are the parameters for these scenarios defined? It would be useful to make a stronger link between the sensitivity analysis exercise presented above and these scenarios. What is the final goal?

l. 643: How "quantitatively"?

l. 684: "sensitivity analysis": to which one do you refer? Global or local sensitivity analysis results? The distinction should be made more explicit for all the scenarios.

---

## Author Comment (AC2) · 28 Jul 2017

Here is a brief summary of the changes made in the manuscript:

1) Shorten and restructure the introduction

2) Revise the elaboration of global sensitivity analysis by Morris method

3) Almost completely rewrite the section of sensitivity result part (section 4) by discussing each major finding separately in the paragraph.Ăă

4) Clearly state that the local sensitivity analysis is necessary to be included, since it presents the parameter sensitivities particularly for modeling seawater intrusion in the

WKP (with the specified values of parameters in WKP). The global sensitivity analysis evaluates the parameter sensitivities within the ranges in general, in addition to the parameter non-linearity and interaction.

5) Completely rewrite the conclusion section with bullet points to highlight the major findings from sensitivities analysis

Again, we would like to appreciate all your comments and suggestions in the revision process.

Thanks Zexuan

---

## Author Response (AR1)

Authors' response:

HESS-2017-85: Xu et al., Numerical modeling and sensitivity analysis of seawater intrusion in a dual-permeability coastal karst aquifer with conduit networks, Hydrol. Earth Syst. Sci. Discuss.,

Anonymous Referee #1

In the submitted paper Xu et al. apply local and global sensitivity analysis on a density driven distributed model (SEAWAT) to simulate a coastal aquifer I Florida (US). They

10  use the knowledge of previous studies to define the boundary conditions and initial parameter sets of the model. Then they apply a local sensitivity analysis on the 11 model parameters in respect to various output variables of the simulated matrix and conduit systems. The same analysis is repeated with a global sensitivity analysis method (Morris) to account for interactions among the model parameters. The parameter describing the

15  salinity at the submarine spring outlet was found to be most sensitive but also the parameters describing the conduit properties were found to influence both the conduit and the matrix behaviour. The results of the more elaborate global sensitivity analysis scheme differ for several model parameters indicating that parameter interactions have to be considered. Finally, the authors use the simulations obtained by the sensitivity analysis to

20  conclude about the sensitivity of the karst system to external changes and about the most gaining observations concerning model parameter identification.

The paper tackles a very interesting field of research, which is the evaluation of distributed models via sensitivity analysis. The authors clearly show that such analysis

25  provides valuable understanding of the model and system and they also highlight that the choice of the sensitivity analysis method has strong impact on the results and conclusions. For those reasons I definitely recommend this paper for publication in Hydrology and Earth System Sciences. However, some weaknesses have to be removed first:

Response: Thanks for the positive evaluation of our work, and we agree that a revision is necessary to improve the quality of the manuscript for publication in HESS. Below are our responses to address the review comments. We would like to further revise the manuscript after the editor makes a decision on this paper.

1. The paper is much too long. In their last paragraph of the conclusions (and also in the abstract) the authors clearly state the main outcomes of their research. However, within the body of the manuscript, they lose themselves in details to often.

40 Response: We agree that it is necessary to reduce the manuscript length. We plan to shorten the introduction section and to remove the redundancy in the result section. The main outcomes of this study will be highlighted in the conclusion part and throughout the manuscript.

45 2. The usage of two sensitivity analysis schemes provides a lot of insights into their differences. However, the authors do not explain why they actually compare them. For many modellers the interaction among parameters is an accepted fact. So for the sake of focus and length of the manuscript: Is it really necessary including the local sensitivity analysis? If not, delete. If yes, provide more explanation why.

Response: We believe it is still necessary to include the local sensitivity analysis in the paper, since it presents the parameter sensitivities in the maximum seawater intrusion case, in which seawater intrudes significantly inland. The authors and readers have special interests in understanding the seawater intrusion under such particular scenario. In
55 addition, local sensitivity study is cheap in computational, and easy to compute.

Comparing the results of local/global sensitivity analysis may help us understand: (1) whether the parameter correlation affect the identification of important parameters, (2) whether the nonlinearity of the model affect the identification of important parameters.
60 The parameter correlation and nonlinear significantly affect the results of sensitivity

analysis, as we see different ranking of important model parameters. Therefore, we need pay special attention to these issues when we conduct model calibration and/or uncertainty analysis.

65  3. Some more links between the model setup and field observations/previous work is necessary. It is clear that a lot of previous work was done at the study site. But sometimes it would be helpful providing some summarizing information in addition to the reference to the previous studies.

70  Response: We thank the reviewer for this suggestion, and will add a more detailed introduction of model setup with its connection to previous studies in this area. We will also add a brief summary of the implications from previous modeling studies in the model setup section. With respect to previous work on sensitivity analysis, Shoemaker et al. (2004) is the only paper that we are aware of for parameter sensitivities in seawater

75  intrusion, although there are several studies about parameter sensitivities and uncertainty quantification in karst aquifer. We plan to include these literatures in this paper during the revision.

4. The elaboration of Morris's method has to be improved.

80

Response: We will refine the elaboration of Morris' method with an improved introduction.

5. A clear discussion relating these results to the result of other is missing.

85

Response: We don't fully understand the reviewer's question. If the reviewer wants us to compare this study with other studies, however, there is no other studies ever addressed the same issue before. In previous answers, we also answered that a more detailed discussion of the previous studies on sensitivity analysis (Shoemaker et al. 2004) will be

90  included in the revised manuscript.

6. No state of the art of sensitivity analysis is missing (and no comparison to other sensitivity analysis studies with lumped of distributed approaches in karst).

Response: We are only aware of several papers on sensitivity analysis for karst aquifers, and will compare the previous studies with this study, although the researches focus on different site and scientific questions. The meaning and benefits of sensitivity analysis of this and other researches will be highlighted in the introduction and conclusion parts to better present the overall objectives of this study.

I think these corrections can all be done within the frame of moderate revisions. Please find some more specific comments in the attached and commented pdf.

Response: We thank the reviewer for the detailed comments, and will address them during the revision.

Anonymous Referee #2

The paper by Xu et al. presents the analysis of results with using local and global sensitivity analysis and different scenarios on a karst coastal aquifer system with seawater intrusion. The topic is potentially very interesting but to be useful to the reader, the paper needs significant improvements. The main concern is about the overall goal of the study. The abstract concentrate mainly on the comparison between local and global sensitivity results, but then a significant portion of the paper deals with scenarios.

From a first reading, it is not clear the purpose of developing all these scenarios and how this connects to the previous sensitivity analysis. From my understanding, the scenarios are developed based on the most sensitive parameters and are expected to show how changes in those parameters can affect the results, but this needs to be more clearly explained.

Response: The reviewer is right about the purpose of having the scenarios in the sensitivity analysis. Based on the sensitivity analysis results, we set up the scenarios by changing the important parameters. The physical meanings of the changing parameters indicate either sea level, precipitation discharge or the length of drought period. For example, the extent of seawater intrusion is quantitatively measured in the scenarios of various sea levels. Although hydraulic conductivity is relatively an important parameter as well, the scenarios of various K are not simulated since it is a constant in a specified study site without change with time. We will attach a clearer explanation in the revised manuscript.

I have serious concerns about how the local sensitivity results are presented: the analysis lacks completely the presentation of the parameter correlation coefficients which are provided as output by UCODE, but are not presented here. CSS without parameter correlation coefficients is not informative and needs to be combined to the correlation analysis.

Response: We respectfully disagree with the reviewer for several reasons. First of all, our local sensitivity analysis is based on CSS, which was defined in Section 2.1 of the manuscript. In addition, if we are not mistaken, CSS does not consider parameter correlation but data correlation. Taking equation (2) of our manuscript (equation 4.5 of the book of Hill and Tiedeman cited in the manuscript), the covariance matric, omega, is for data, not for parameters. In fact, the conventional local sensitivity analysis (e.g., the one used in UCODE and also in this study) does not address parameter correlation. This was part of the reason of conductivity the Morris-based global sensitivity analysis.

Furthermore, there is not good explanation of which observations are used in this analysis. There should be heads and salinity field observations, but it is not clear how many, where and mostly which is the weight which is applied to each observation.

Response: When conducting sensitivity analysis, we do not need to have "field observations". We only need to know which "model simulations" are sensitivity to which parameters. We can have many model simulations. Very little head and salinity observations can be used in this analysis, since it is extremely difficult to collect water samples or install sensors in the subsurface karst conduits (100 meter deep). Xu et al. (2016) summarized the existed field observations within the conduit system. However, these measurements are far inland and not applicable to be used for model calibration in this study.

How are the local sensitivity analysis results used to build the simulations for global sensitivity analysis? Are the same parameters included in both the analyses? Which are the key additional information that we are getting from this double analysis? Often local sensitivity results can be used to discriminate which parameters should be included in the global sensitivity analysis, but this does not seem to be the case in this paper. The two analyses seem to be performed exactly on the same set of parameters.

Response: We did not use the local sensitivity results to build the simulations for global sensitivity analysis. Both the local and global sensitivity analyses evaluate the same parameters. However, the global analysis evaluates parameters within a certain range, while the local analysis only evaluates the specified parameter values specifically for the study region (WKP). The key additional information in global sensitivity analysis is the assessments of parameter sensitivities within the range and the parameter interaction/correlation with each other.

The introduction is too long and does not get exactly to the point of the paper. Sensitivity analysis and calibration of similar models have already been performed and it would be good to point out where. The discussion of the sensitivity analysis results and of the different scenarios should be organized and made more concise. There are also some inconsistencies with the parameter names that should be checked. My suggestion would

be to develop a summary table with all the results for both sensitivity and again for the
scenarios to help the reader understanding which are the key findings of this analysis.

Response: We thank the reviewer for the excellent suggestion. We will shorten the
introduction, link this research with previous sensitivity studies in karst, and highlight the
overall objective of this study. The result and discussion parts will be re-written and re-
organized. A summary with bullet points are made in the conclusion section to highlight
the major findings in this study.

Some more specific comments include:
l.48-49: are the parameters mentioned here calibrated parameters?

Response: These are the parameters and boundary conditions adjusted in the scenarios
simulations. All the parameters of aquifer characteristics are calibrated.

l. 132: Which are the issues with VDFST-CFP? Why cannot be used in this case?
Without explanation it is not clear why this other method should be mentioned.

Response: VDFST-CFP is not able to hand the issue addressed in this paper, because of
computational issues related to the aquifer geometry and the scale of spatial domain. The
numerical model in this study is setup to study seawater intrusion in a field scale, with the
real dimension and parameter values of the porous medium and the conduit in the aquifer.
The current version VDFST-CFP is not able to simulate groundwater flow and solute
transport within a large conduit network, because the discrete-continuum modeling
method requires the cell size to be at least one order of magnitude larger than the conduit
diameter. The computational cost is another constraint. For more details, please refer to
Xu and Hu (2017, WRR). We will add a brief explanation in the manuscript to explain
why VDFST-CFP is not applicable to this study.

L. 207: please spell out EEs

210 Response: EEs is the abbreviation for element effects, which was spelled out above equation (4) of the manuscript. The reviewer probably missed the spell-out.

l. 287: the goal here seems to be just performing sensitivity analysis and not calibration. But for running the scenarios, it should be clarified how the values of the parameters have
215 been selected. After which type of calibration? Or the values calibrated in previous studies have been used?

Response: While the reviewer is correct that calibration is always performed to obtain the nominal values used for local sensitivity analysis, we want to point out that the nominal
220 values can be selected based on literature data without conducting model calibration. This may be a limitation for the local sensitivity analysis, because it is not an easy task to selecting nominal parameter values. This problem can be resolved by conducting a global sensitivity analysis, which does not require nominal parameters values but parameter ranges (the ranges can be easily determined based on a literature review). In this study,
225 for the local sensitivity analysis, we did not conduct model calibration, since there is no observational data available to calibrate the model, especially the simulations within the conduit network. The parameter values of aquifer properties (hydraulic conductivity, effective porosity, specific storage, and dispersion coefficient) used in this study are based on a literature review of previous modeling studies in this study area.  Information
230 on this regard was provided in section 3.1, but we will provide more detailed information to clarify in the revised manuscript.

l. 453: why the arithmetic mean of CSS?

235 Response: Here we mean the average of CSS values in all evaluated locations (cell #25 - #75 along the conduit/porous medium layer).

l.470: this was already said at line 455. The discussion should be more compact and better organized.

240

Response: We will delete the redundant sentence.

l. 568: "groundwater seepage velocity" : are these used as observations?

245     Response: Groundwater seepage velocity is not used as observations or model simulation in this study. We actually don't have any observation in this study. See the answers above.

l. 610: clarify which are the field observations and how they have been weighted

250     Response: We will clarify in the revised manuscript that we did not use any observations for the sensitivity analysis in this paper. As explained above, sensitivity analysis does not require observations. This paragraph is talking about the implication and significance of sensitivity analysis on understanding seawater intrusion in karst aquifer. This study points out that the head and salinity field observations and numerical simulations within the

255     conduit network are particularly important.

l. 619-621: check the English

Response: We will break this long sentence.

260

l. 626: the DSP package was not mentioned earlier

Response: DSP means the dispersion package in SEAWAT. We tried to deactivate the DSP package to verify that dispersion is negligible within the conduit network in this

265     study. We will clarify this in the revised manuscript

l. 630: local sensitivity allows as well to understand interactions and correlations between parameters but it is not presented here

270    Response: We respectfully disagree with the review that the conventional local sensitivity analysis used in this study considers parameter interaction and correlation. Because the local sensitivity is based on the first order derivative of a model simulation with respect to a model parameter, it does not consider parameter interaction/correlation. We would appreciate, if the review could provide some references in which local sensitivity analysis

275    address parameter interaction/correlation.

Chapter 5: how are the parameters for these scenarios defined? It would be useful to make a stronger link between the sensitivity analysis exercise presented above and these scenarios. What is the final goal?

280

Response: We thank the reviewer for the excellent suggestion for linking between the sensitivity analysis and the scenarios, and this will be added in the revised manuscript. The goal of having these scenarios is to quantitatively evaluate the distance of seawater intrusion within conduit and its effects on the surrounding porous medium, under

285    different scenarios. The distance between the shoreline and the interface of seawater and freshwater mixing is measured as the quantity for evaluating seawater intrusion.

l. 643: How "quantitatively"?

290    Response: Please see the answer above.

l. 684: "sensitivity analysis": to which one do you refer? Global or local sensitivity analysis results? The distinction should be made more explicit for all the scenarios.

295    Response: We mean both local and global sensitivity analysis results. We will clarify this in the revised manuscript.

findings from sensitivities analysis

[revised manuscript text omitted]

conduit network that determine the extent of seawater intrusion within the conduit. However,

other parameters are relatively less important with small mean and standard deviation of EEs.

| Page 36: [16] Deleted | Zexuan Xu | 7/7/17 4:52:00 PM |

Different from salinity simulation, Although ssalinity at the submarine spring (SC) no longer has

does not have the largest mean of EEs but is still, it is still an important parameter for head

simulation. However, parameter SC still has the  with a large standard deviation of EEs, since the

derivatives vary in different evaluated locations, with due to its non-linear relationship to head

simulation shown in (Fig. 8). The parameter SC, indicates that the hHead simulations are also

sensitive to the boundary conditions of salinity in the transport model, because equivalent

freshwater head is a function of density in terms of salinity simultaneously in the coupled

variable-density flow and transport model for simulating seawater intrusion.

| Page 36: [17] Deleted | Zexuan Xu | 7/7/17 5:14:00 PM |

for effective porosity and specific storage of the conduit

| Page 36: [18] Deleted | Zexuan Xu | 7/4/17 12:46:00 PM |

 are much larger than hydraulic conductivity (HY_C), indicating that head simulation in the

conduit is more sensitive to effective porosity and specific storage, rather than the hydraulic

conductivity

| Page 36: [19] Deleted | Zexuan Xu | 7/4/17 12:50:00 PM |
|---|---|---|

The uncertainty of non-laminar flow rate calculation in the continuum model is highlighted in the sensitivity analysis.

| Page 36: [20] Deleted | Zexuan Xu | 7/4/17 1:08:00 PM |
|---|---|---|

, which means that the Darcy equation in SEAWAT model

| Page 36: [21] Deleted | Zexuan Xu | 7/4/17 1:15:00 PM |
|---|---|---|

As a result, the sensitivities of hydraulic conductivity have significant uncertainty to evaluate the permeability of conduit system under non-laminar flow condition.

| Page 36: [22] Formatted | Zexuan Xu | 7/4/17 1:11:00 PM |
|---|---|---|

Font:(Asian) +Theme Body Asian (宋体)

| Page 36: [23] Deleted | Zexuan Xu | 7/5/17 11:19:00 AM |
|---|---|---|

The identified by the means and standard deviations of EEs

| Page 36: [24] Deleted | Zexuan Xu | 7/5/17 11:17:00 AM |
|---|---|---|

. The mean and standard deviation values indicate that parameters HY_P and SC are the two most important parameters for simulations in the porous medium

| Page 36: [25] Deleted | Zexuan Xu | 7/11/17 9:03:00 AM |
|---|---|---|

Porous medium h Hydraulic conductivity of the porous medium (HY_P) is an important term in the flow equation for solving head and groundwater seepage velocity in the flow equation, which then determines and the advective velocity of for the transport equation in the porous medium. As the boundary condition of conduit system, salinity at the submarine spring (SC) is also important as the major source of seawater intrusion) determines the equivalent freshwater head at the inlet of seawater intrusion and affects simulations not only within the conduit system, but also in the surrounding porous medium via exchanges between the two domainssystems. Similar to the largest CSS value of parameter SC in the local sensitivity result, tThe global sensitivity

analysis result of parameter SC highlights the significance of the interaction between the conduit

and the porous medium domains in a dual-permeability aquifer. However, salinity at the

submarine spring (SC) with respect to simulations in the porous medium has the largest CSS in

the local sensitivity study, while t
* * *
**Page 36: [26] Deleted**       **Zexuan Xu**       **7/26/17 11:59:00 AM**

than the CSS of hydraulic conductivity of the porous medium (HY_P) is has much smaller CSS

value in the local sensitivity result  (Fig. 6), buthowever, and also has larger mean of EEs in the

global sensitivity study than parameter SC
* * *
**Page 38: [27] Deleted**       **Zexuan Xu**       **7/7/17 5:38:00 PM**

In general, parameter sensitivities of simulations in the porous medium are more

complicated than those in the conduit, especially for the head simulations. The global sensitivity

analysis provides an insight of the parameter interactions and the higher-order relationship with

respect to simulations.
* * *
**Page 39: [28] Deleted**       **Zexuan Xu**       **7/6/17 2:29:00 PM**

are simulated by the SEAWAT model and quantitatively measured and measuredanalyzed
* * *
**Page 39: [29] Deleted**       **Zexuan Xu**       **7/6/17 2:31:00 PM**

In addition, the length of elapsed time in simulation is constant in the sensitivity analysis for

consistent comparison purposes. In Sect. 5.4, tThe extents of seawater intrusion in a coastal karst

aquifer duringunder an extended low rainfall periods conditions are evaluated by extending the

elapsed time in simulation in Sect. 5.4, although which is constant in the sensitivity analysis for

consistent comparison purposes.
* * *
**Page 39: [30] Deleted**       **Zexuan Xu**       **7/6/17 11:27:00 AM**

are evaluated at the specified parameter values in this case
* * *
**Page 39: [31] Deleted**       **Zexuan Xu**       **7/6/17 11:42:00 AM**

This case is also set as the benchmark for the following scenarios.
* * *
**Page 45: [32] Deleted**          **Zexuan Xu**          **7/6/17 6:13:00 PM**
* * *
Salinity at the submarine spring (SC) is identified as the most important parameter to the simulations in both two domains, because the submarine spring is the major entrance of seawater intrusion into the conduit as pathway in the aquifer. The boundary conditions and hydrological characteristics of the conduit, including sea level at the submarine spring (H_SL), hydraulic conductivity (HY_C) and effective porosity (PO_C) are important to the simulations in the conduit as well. On the other hand, the simulations in the porous medium also are sensitive to the boundary conditions (SC and H_SL) and hydrological characteristics of the conduit, such as specific storage and effective porosity (SS_C and PO_C), due to the interaction and exchange between the two domains. Sensitivity analysis indicates that the observations and simulations in the conduit are especially important for understanding hydrogeological processes and modeling seawater intrusion in such a dual-permeability karst aquifer. In addition, the largest CSS values of parameter sensitivities can be found around the mixing zone position. The local sensitivity analysis in this study confirms the conclusions of sensitivity studies in a homogeneous aquifer in Shoemaker (2004), also highlights the values of conduit network in modeling seawater intrusion in a coastal karst aquifer.

The global sensitivity analysis indicates that head simulation in the conduit is more sensitive to effective porosity (PO_C) and specific storage of the conduit (SS_C), instead of hydraulic conductivity. The conduit flow easily becomes non-laminar and beyond the capability of Darcy equation in SEAWAT model, which assumes a linear relationship between specific discharge and head gradient. Therefore, the uncertainty of conduit permeability is difficult to be accurately evaluated by hydraulic conductivity in the continuum model. Different from the local sensitivity study, hydraulic conductivity of the porous medium (HY_P) has the largest mean of

EEs with respect to head simulation in the porous medium Simulations are non-linear to parameter SC, which has the largest derivative in the specified evaluated location in the local sensitivity study., Dispersivity is no longer an important parameter for simulations in the conduit, which is different from Shoemaker (2004), because advection is dominated in the solution of saline water transport with turbulent flow in the conduit, as well as the relatively fast seepage flow in the surrounding porous medium. In the salinity profile, the mixing is mostly due to numerical dispersion instead of the solution of dispersion equation, since Peclet number is extremely large and beyond the criteria of solving transport equation by finite difference method.
* * *
**Page 45: [33] Moved to page 45 (Move #5)      Zexuan Xu                          7/6/17 9:22:00 PM**

Seawater intrudes significantly further landward through the conduit, and flows into the surrounding porous medium via the exchange on the pipe wall.

---

## Referee Report (RR1)

The paper has significantly improved in clarity. My impression is that the discussion of the results is pretty long and articulated and would benefit a lot from a summary table or even a summary as bullet points. It is still difficult to follow it.

I am providing below some comments on the response letter and some minor comments on the nw manuscript.

Line 137 in the response letter: I believe there is some misunderstanding regarding local sensitivity analysis and how they should be used. The weighting is based on the variance-covariance matrix of the data errors. This is different than the correlation matrix of the parameters. The answer of the authors regarding omega is not correct.

The CSS uses the weighting, so it only accounts for the variance-covariance of the data errors. I think this is the point of the authors, and I agree – but this is only because CSS by themselves do not tell the whole story and need to be used combined with other measures. To get a bigger picture, it is important to consider them with the parameter correlations, which UCODE does calculate. UCODE calculates also leverage that is another measure including information on correlations.

Their final statement that says the conventional local sensitivity analysis (e.g., the one used in UCODE and also in this study) does not address parameter correlation is not correct. UCODE does calculate the variance-covariance matrix for the parameters and also the correlations. Whether it is chosen to use them or ignore them, this is the authors' choice. It is not a shortcoming of UCODE, but of the way it is used. Relying only on CSS, does not allow getting the full picture. It is true that user needs to make the interpretations by considering these together, so the problem is NOT "addressed" automatically. Guidelines on how to use CSS, parameter correlation coefficients and also leverage at the same time are provided in Hill and Tiedeman, 2007. The most informative way of going through all the measure is following the sensitivity analysis developed for the synthetic exercise presented in the book. I can provide more references upon request.

Line 270 in the response letter: same comment as above

In the new manuscript:

Line 31: What does WKP stand for?

Line 30-35: the justification of why Local and Global analysis are performed is very weak

Line 268-270: check the sentence

Line 382: parameters eventually to be adjusted during calibration

Line 600: it would be good to understand the connection between the sensitivity analysis and this has been used to design scenarios

Line 601: remove one "the"

Line 742: any idea of why this is happening?

---

## Author Response (AR2)

Authors' responds:
HESS-2017-85: Xu et al., Numerical modeling and sensitivity analysis of seawater intrusion in a dual-permeability coastal karst aquifer with conduit networks, Hydrol. Earth Syst. Sci. Discuss.,

Editor:

In particular, I expect that the Authors improve the paper, by taking into account and properly answering both the reviewers' comments and the questions that I list below.
1) The presentation and the language should be improved. I provide several suggestions in the attached marked version of the manuscript, where some of the most obscure (if not incorrect) sentences are annotated.

Response: Thanks for providing the important suggestions in the languages. We have corrected the grammar issues and rephrase the sentences. The Morris methodology and global sensitivity analysis results sections are rewritten to address some issues from the reviewers.

2) From the physical point of view the choice of a 2D model is not supported by solid arguments. In fact, a 2D model assumes that the quantities are constant along the direction perpendicular to the modelled cross section, i.e., parallel to the coast line. Is this realistic? Does the conduit network have a large extension along that direction? How are the results affected from 3D flow and transport in the porous matrix?

Response: We admit the 2D model has some limitations on simulating the seawater intrusion in the entire aquifer. The conduit network does not have a large extension parallel to the coastline, however, relatively high hydraulic conductivity layers are found at nearly at the same depth as the conduit network. Overall, we only try to simulate the cross section that containing the conduit segment, not the other parts of the aquifer. The editor is right that the exchanges between the conduit and surrounding porous medium are computed in this 2D model, however, the effects of flow and transport along the direction perpendicular to the cross section is ignored in this study.

3) I fully agree with the remark by Referee #2 that the discussion of the results is very long and difficult to follow.

Response: The discussion of global sensitivity analysis result is shortened. We delete the redundant results which has been presented in the previous local sensitivity analysis subsection, and only discussed the important parameters and major findings in this study.

4) I fully agree also with the remark by Referee #2 about CSS and weighting.

Response: We have clarified the CSS and weighting in the responses to Referee #2 (see below) and also in the revised manuscript.

Anonymous Referee #1

With the resubmitted version of their manuscript, Xu et al provide a significantly improved version of their work. The manuscript was shortened to add more focus. They provide a sound explanation why they chose to apply 2 methods for sensitivity analysis. A better link between previous research at the study site and this study was established and the elaboration of parts of the sensitivity analysis methods was improved.

I agree with the authors that parameter correlation does not have to be addressed in the frame of the local sensitivity analysis as (1) local parameter analysis is by definition very limited in identifying parameter interactions and (2) parameter interactions are explicitly addressed by Morris's method, which is by definition a "global" parameter sensitivity analysis scheme.

Some confusion came up addressing my comment on including more comparison to the work of others. Indeed, this was partially addressed by including Shoemaker et al. (2004) in the discussion but the link to studies that used sensitivity in karst modelling is still weak. The authors indicate that they are aware of just few karst studies using sensitivity analysis but a quick scan of the literature revealed couple of studies using local (Oehlmann et al., 2014), regional (Chang et al., 2017; Hartmann et al., 2015) and global sensitivity analysis (Chen et al., 2017; Hartmann et al., 2013) in karst modelling for both lumped and distributed modelling approaches.

Including some of these (or similar) references to their manuscript will strengthen the state of the art of karst sensitivity analysis as well as the discussion. After these very minor suggestions have been applied I would feel very confident recommending this study for publication in Hydrology and Earth System Sciences.

Response: Thanks for providing these related references. We agree with the reviewer and include these latest references in the introduction section to provide a more complete description of karst sensitivity analysis.

Anonymous Referee #2

The paper has significantly improved in clarity. My impression is that the discussion of the results is pretty long and articulated and would benefit a lot from a summary table or even a summary as bullet points. It is still difficult to follow it.

Response: We have shortened the discussion part, particular the global sensitivity analysis subsection, by deleting the redundant discussion that has been addressed in the local sensitivity analysis results. The major findings in this study are summarized as bullet points in the conclusion section.

I am providing below some comments on the response letter and some minor comments on the new manuscript.

Line 137 in the response letter: I believe there is some misunderstanding regarding local sensitivity analysis and how they should be used. The weighting is based on the variance-covariance matrix of the data errors. This is different than the correlation matrix of the parameters. The answer of the authors regarding omega is not correct.

Response: We agree with the reviewer here. We have clarified that the omega is set as the variable-covariance matrix of parameters as the data errors in the revised manuscript.

The CSS uses the weighting, so it only accounts for the variance-covariance of the data errors. I think this is the point of the authors, and I agree – but this is only because CSS by themselves do not tell the whole story and need to be used combined with other measures. To get a bigger picture, it is important to consider them with the parameter correlations, which UCODE does calculate. UCODE calculates also leverage that is another measure including information on correlations.

Response: We agree with the reviewer that UCODE computes the parameter correlations, and CSS does not consider parameter correlation but data correlation. We add the parameter correlations computed from UCODE and discuss the meaning of correlations in subsection 4.1.3 in the revised manuscript.

Their final statement that says the conventional local sensitivity analysis (e.g., the one used in UCODE and also in this study) does not address parameter correlation is not correct. UCODE does calculate the variance-covariance matrix for the parameters and also the correlations. Whether it is chosen to use them or ignore them, this is the authors' choice. It is not a shortcoming of UCODE, but of the way it is used. Relying only on CSS, does not allow getting the full picture. It is true that user needs to make the interpretations by considering these together, so the problem is NOT "addressed" automatically. Guidelines on how to use CSS, parameter correlation coefficients and also leverage at the same time are provided in Hill and Tiedeman, 2007. The most informative way of going through all the measure is following the sensitivity analysis developed for the synthetic exercise presented in the book. I can provide more references upon request.

Response: We agree that the CSS only does not provide a full picture of parameter sensitivities. We add the parameter correlations from UCODE in this revised manuscript.

Line 270 in the response letter: same comment as above
In the new manuscript:
Line 31: What does WKP stand for?

Response: WKP stands for the Woodville Karst Plain. We change the wording in the revised manuscript.

Line 30-35: the justification of why Local and Global analysis are performed is very weak

Response: In the revised manuscript, we have clarified the objectives of this study and the importance of sensitivities analysis in the last two paragraphs of the introduction section.

140 Line 268-270: check the sentence

Response: Revised.

Line 382: parameters eventually to be adjusted during calibration

145

Response: I'm not sure if I understand the reviewer's question correctly. Obviously, the parameters are adjusted during calibration to compute the salinity and head simulations. However, we didn't do model calibration in this study due to the lack of observational data. The parameters (actually, boundary conditions) to be adjusted in the scenario
150 analysis is determined by the local sensitivity analysis.

Line 600: it would be good to understand the connection between the sensitivity analysis and this has been used to design scenarios

155 Response: The local sensitivity analysis results are used to design the scenarios. The two most important parameters and boundary conditions identified in local sensitivity analysis, including the salinity and sea level at the submarine spring, are evaluated in the scenarios. The scenarios also use all the parameter values evaluated in the local sensitivity analysis. We have clarified the connection in section 5.

160
Line 601: remove one "the"

Response: Revised.

165 Line 742: any idea of why this is happening?

Response: The size of grid cells and advective velocity are two major factors of numerical dispersion, which can be reduced by decreasing the grid cell size (increasing the spatial resolution) and/or decreasing the advective velocity. In this study, the
170 advective velocity is relatively fast associated with the large hydraulic conductivity. On the other hand, the computational time/cost will be a trouble for sensitivity analysis when increasing spatial resolution, since the model will run many times. We add some explanation in the last paragraph of conclusion section.

[revised manuscript text omitted]

---

## Author Response (AR4)

Authors' responds:
HESS-2017-85: Xu et al., Numerical modeling and sensitivity analysis of seawater intrusion in a dual-permeability coastal karst aquifer with conduit networks, Hydrol. Earth Syst. Sci. Discuss.,

Editor:

The paper has been revised, by including some information and comments, following the comments to the last version. Unfortunately, two basic scientific concerns remain.
1) About the use of a 2D model, I do not agree with the sentence at lines 280 to 281 ("The exchange fluxes... are ignored"). In particular, I cannot find a proof of this statement in the paper and I simply suggest to erase the sentence.

Response: Erased. The paragraph of 2D model limitation has been rewritten.

2) The weight appearing in equation (2) is "set as 1.0 equally for the 11 evaluated locations (column #25 to #75 with an interval of 5 cells) for salinity and head simulations in this study" (see lines 172 to 174). In other words the weight omega_{i,i} for head simulations is 1 m^2 and 1 PSU^2, respectively for head and salinity. I recall that head
varies between 0 m and 1.52 m in the simulated domain, whereas salinity varies between 0 PSU and 35 PSU. Therefore the weight does not take into account the difference in measurement units and the variability of the two quantities. In other words, with this choice (same weight for head and salinity simulations) dss is not really "dimensionless", because it does not take into account the different measurement units and therefore the
different numerical values of head and salinity. This choice must be physically-based and explained, otherwise the scientific merit of the paper will be significantly lowered.

Response: We apologized for the unclear statement and any misunderstanding in the previous revision. We have carefully recalled our UCODE sensitivity analysis
methodology and double-checked input files, since it has been done a while ago. Briefly speaking, the weights of head and salinity are not the same and computed by the error variances/standard deviations in this study.

In the previous revision, we would like to explain that the weights for the 11 evaluated
locations from column #25 to #75 are equal, for head and salinity simulation, respectively. In fact, the weights of DSS for head and salinity simulations are computed by the inverse of error variances (square of error standard deviation) as $\omega_{ii} = 1/\sigma^2$. The values of error standard deviation used in this study are referenced from Shoemaker et al. (2004). The head measurement error was assumed to be normally distributed with a mean
of zero and a standard deviation of ~0.003 m (originally as 0.01 ft in the UCODE input files). The standard deviation was based on standard error estimates for water levels measured in wells by the USGS in southern Florida. The salinity measurement error was assumed to be normally distributed with a mean of zero and a standard deviation of ~0.1 PSU (kg/m^3). Salinity range from 3 to 35 PSU, and for this range using a 0.1 PSU

standard deviation was thought to be appropriate based on discussion with USGS water
     quality personnel (Shoemaker et al., 2004).

     We have revised the manuscript to clearly explain the weights calculations. The editor is
     right that different measurement units for head and salinity should be taken into account
and have be addressed in the dimensionless scaled sensitivities computation.

     A technical remark. I recall that SI measurement units should be used throughout the
     whole paper, whereas other units still appear somewhere in the text and in many figures.

Response: Thanks for pointing out this. We have changed the non-SI units in the figure
     captions and re-plot the figures.

[revised manuscript text omitted]